# HOW TO TRAIN RNNS ON CHAOTIC DATA?

## ABSTRACT

Recurrent neural networks (RNNs) are wide-spread machine learning tools for modeling sequential and time series data. They are notoriously hard to train because their loss gradients backpropagated in time tend to saturate or diverge during training. This is known as the exploding and vanishing gradient problem. Previous solutions to this issue either built on rather complicated, purpose-engineered architectures with gated memory buffers, or - more recently - imposed constraints that ensure convergence to a fixed point or restrict (the eigenspectrum of) the recurrence matrix. Such constraints, however, convey severe limitations on the expressivity of the RNN. Essential intrinsic dynamics such as multistability or chaos are disabled. This is inherently at disaccord with the chaotic nature of many, if not most, time series encountered in nature and society. Here we offer a comprehensive theoretical treatment of this problem by relating the loss gradients during RNN training to the Lyapunov spectrum of RNN-generated orbits. We mathematically prove that RNNs producing stable equilibrium or cyclic behavior have bounded gradients, whereas the gradients of RNNs with chaotic dynamics always diverge. Based on these analyses and insights, we adapt the old idea of teacher forcing to yield an effective yet simple training technique for chaotic data, and offer guidance on how to choose relevant hyperparameters according to the Lyapunov spectrum.

## 1 INTRODUCTION

Recurrent neural networks (RNNs) are widely used across various fields in engineering and science for learning sequential tasks or modeling and predicting time series (Lipton et al., 2015). Yet, they struggle when long-term temporal dependencies, very slow, or hugely varying time scales are involved (Hochreiter, 1991; Bengio et al., 1994; Schmidt et al., 2021; Li et al., 2018; Rusch & Mishra, 2021a). Time series or sequential data with such properties are, however, very common in fields like climate physics (Thomson, 1990), neuroscience (Fusi et al., 2007; Russo & Durstewitz, 2017), ecology (Turchin & Taylor, 1992), or language processing (Cho et al., 2014b). Training RNNs on such data is hard because the loss gradients backpropagated in time easily saturate or diverge in this process. This is commonly referred to as the exploding and vanishing gradient problem (EVGP) (Hochreiter, 1991; Bengio et al., 1994; Pascanu et al., 2013).

One solution to the EVGP is based on specifically designed RNN architectures with gating mechanisms, such as long short-term memory (LSTM) (Hochreiter & Schmidhuber, 1997) or gated recurrent units (GRU) (Cho et al., 2014a). These architectures allow states at earlier time steps to more easily influence activity much later through a kind of protected memory buffer, thus alleviating the EVGP by structural design. In practice, such models need to be backed up by further techniques like gradient clipping to keep the gradients in check (Pascanu et al., 2013). The relatively complex architectural design of these networks impedes their mathematical analysis and requires reverse engineering after training (Maheswaranathan et al., 2019; Monfared & Durstewitz, 2020a;b; Schmidt et al., 2021). Partly to forego these complications, a variety of other solutions has been proposed recently, imposing restrictions on the recurrence matrix to bound the gradients (Arjovsky et al., 2016; Chang et al., 2019), or enforcing global stability by design or regularization (Erichson et al., 2021; Kolter & Manek, 2019). Often these procedures dramatically curtail the expressivity of the RNN (Kerg et al., 2019; Orhan & Pitkow, 2020; Schmidt et al., 2021); in particular, they rule out chaotic dynamics (see further discussion below).

This is at odds with the plethora of chaotic phenomena in nature, engineering, and society. Chaotic dynamics are commonplace, almost default in any complex physical or biological system. This includes scientific areas as diverse as neuroscience (Durstewitz & Gabriel, 2007; van Vreeswijk & Sompolinsky, 1996), physiology (Kesmia et al., 2020), geophysics (Sivakumar, 2004), climate systems (Tziperman et al., 1997), astrophysics (Laskar & Robutel, 1993), ecology (Duarte et al., 2010), chemical reactions (Field & Györgyi, 1993), cell (Olsen & Degn, 1977) or population (May, 1987) biology. Chaotic phenomena are also crucial for the understanding of societal and epidemiological processes, such as the spread of diseases (Mangiarotti et al., 2020), or in economics (Faggini, 2014). They are further relevant in purely technical contexts such as electrical engineering (Tchitnga et al., 2019; Kamdjeu Kengne et al., 2021) or laser optics (Kantz et al., 1993). They have even been suggested to play an up to now largely neglected, but potentially very significant role in speech recognition (Sabanal & Nakagawa, 1996) and natural language processing (Inoue et al., 2021). Hence, in almost any practical setting, chaotic phenomena abound. They cannot, in general, be ignored when devising RNN training algorithms.

Here we offer a comprehensive theoretical treatment of the relation between RNN dynamics and the behavior of the loss gradients during training. We find a close connection between an RNN's loss gradients and the largest Lyapunov exponent of its freely generated orbits. We mathematically prove that RNNs producing stable fixed point or cyclic behavior have bounded gradients. Crucially, however, the loss gradients of RNNs producing chaotic dynamics always diverge. Hence, the chaotic nature of many time series data induces a *principle* problem, and, despite significant efforts in the past to solve the EVGP, training RNNs on such data remains an open issue. We illustrate the implications of our theory for RNN training on several simulated and empirical chaotic time series, and adapt the old idea of *sparsely forced BPTT* as a simple yet effective remedy that enables to learn the underlying dynamics despite exploding gradients.

## 2 RELATED WORKS

*Exploding and vanishing gradients.* While 'classical' remedies of the EVGP (Hochreiter, 1991; Bengio et al., 1994; Pascanu et al., 2013) rest on purpose-tailored architectures with gating mechanisms, which safeguard information flow across longer temporal distances (Hochreiter & Schmidhuber, 1997; Cho et al., 2014a), the focus has recently shifted to simpler RNNs that address the EVGP by restricting the recurrence matrix to be orthogonal (Henaff et al., 2016; Helfrich et al., 2018; Jing et al., 2019), unitary (Arjovsky et al., 2016), or antisymmetric (Chang et al., 2019), or by ensuring globally stable fixed point solutions (Kag et al., 2020; Kag & Saligrama, 2021a), for example through co-trained Lyapunov functions (Kolter & Manek, 2019). However, all these approaches impose strong limitations on the dynamical repertoire of the RNN, enforcing global convergence to fixed points or simple cycles.[1] In doing so, they drastically reduce the expressiveness of these models (Kerg et al., 2019; Orhan & Pitkow, 2020). To address this problem, Erichson et al. (2021) somewhat relaxed the constraints on the recurrence matrix by introducing a skew-symmetric decomposition combined with a Lipschitz condition on the activation function. Another recent approach discretizes oscillator ODEs to arrive at a stable system of coupled (Rusch & Mishra, 2021a) or independent (Rusch & Mishra, 2021b) oscillators which increase the RNN's expressiveness while bounding its gradients. By design (and as acknowledged by the authors), neither of these architectures is capable of producing chaotic dynamics, however, as the underlying ODEs do not allow for exponential divergence of close-by trajectories (a prerequisite for chaos). Given these often principle limitations of parametrically or dynamically strongly constrained models, a fruitful direction may be to modify the training process itself, e.g. through modified or auxiliary loss functions (Trinh et al., 2018; Schmidt et al., 2021), or special procedures for parameter updating (Kag & Saligrama, 2021b) or loss truncation (Williams & Zipser, 1989; Menick et al., 2021). Our empirical evaluation will follow up on such ideas, but also highlight that simple loss truncation, windowing, or architectural solutions like LSTMs are not sufficient.

*Learning dynamical systems.* Surprisingly disconnected from the work on the EVGP and learning long-term dependencies, a huge and long-standing literature deals with training RNNs on nonlinear dynamical systems (DS) (Pearlmutter, 1990; Trischler & D'Eleuterio, 2016; Vlachas et al., 2020), including chaotic systems like the famous Lorenz equations (Lorenz, 1963) or chaotic turbulence in fluid dynamics (Li et al., 2021). Teacher forcing (TF; Williams & Zipser (1989), Pearlmutter (1990),

---

[1]We make this point more formal in Appx. A.1.6.

Doya (1992), see also Goodfellow et al. (2016)) is one of the earliest techniques introduced to keep RNN trajectories on track while training. The idea behind TF is to simply replace RNN states by observations when available, thereby also effectively cutting off the gradients. TF essentially derives from ideas in dynamical control theory, and adaptive schemes that increasingly hand over control to the RNN throughout training have been devised (Abarbanel, 2013; Abarbanel et al., 2018; Bengio et al., 2015). A related technique from the control theory literature is "multiple shooting" (Voss et al., 2004): Here the whole observed time series is chopped into chunks, and for each chunk of trajectory a new initial condition is estimated. Explicit constraints ensure continuity between the separate trajectory bits during optimization. State space models and the Expectation-Maximization algorithm became popular particularly in the 90es for uncovering the latent dynamics underlying a set of time series observations (Ghahramani & Roweis, 1999), and remain an important tool until today (Durstewitz, 2017; Koppe et al., 2019). Most recently, approaches based on variational inference and the reparameterization trick (Kingma & Welling, 2014), like sequential variational autoencoders (SVAE), gained in popularity for DS approximation (Hernandez et al., 2020). "Deterministic" RNNs (i.e., with latent states not treated as random variables), like conventional LSTMs (Vlachas et al., 2018), remain top choices for DS reconstruction, however.

Although connections between DS ideas and loss gradients have been drawn early on (Bengio et al., 1994), so far only particular scenarios (like fixed point attractors) have been considered. Closest to our work is recent work by Schmidt et al. (2021), where non-divergence of loss gradients is established when RNNs converge to fixed points or cycles. However, this was done only for the particular class of piecewise-linear RNNs (PLRNNs), more restrictive conditions for cycles were imposed than assumed here, and - most importantly - the chaotic case on which we focus here was not considered. Another recent study (Engelken et al., 2020) also points out the general connections between Lyapunov exponents and loss gradients that we develop in sect. 3.1, but does not provide any in-depth theoretical treatment, proofs, or empirical evaluation, as we do here. Thus, a systematic theoretical framework that relates RNN dynamics more generally, and across a range of different RNN architectures, to the behavior of its training gradients, is still lacking so far.

## 3 THEORETICAL ANALYSIS: RELATION BETWEEN RNN DYNAMICS AND LOSS GRADIENTS

In our analysis, we will cover all major types of system dynamics (fixed points, cycles, chaos, and quasi-periodicity), and mathematically investigate their implications for the loss gradients. We will do this for all major classes of RNNs, including standard RNNs with largely arbitrary activation function, LSTMs, GRUs, and PLRNNs. The next section will first develop and illustrate the basic intuition behind the relations between RNN dynamics and loss gradients.

### 3.1 PRELIMINARIES: RNN DYNAMICS AND LOSS GRADIENTS

Formally, all popular RNN architectures, including LSTMs, GRUs, or PLRNNs, are discrete time DS, defined by a (first-order-Markovian) recursive prescription for the temporal evolution of the latent states $\boldsymbol{z}_t \in \mathbb{R}^M$ of the form

$$\boldsymbol{z}_t = F_{\boldsymbol{\theta}}(\boldsymbol{z}_{t-1}, \boldsymbol{s}_t), \tag{1}$$

where $\boldsymbol{s}_t \in \mathbb{R}^N$ is the input at time $t$ and $\boldsymbol{\theta}$ are RNN parameters. For instance, for standard RNNs we have $F_{\boldsymbol{\theta}}(\boldsymbol{z}_{t-1}, \boldsymbol{s}_t) = f(\boldsymbol{W}\boldsymbol{z}_{t-1} + \boldsymbol{B}\boldsymbol{s}_t + \boldsymbol{h})$, where $f$ is an element-wise activation function like $\tanh$ or a rectified linear unit (ReLU).

Assuming we start at some initial value $\boldsymbol{z}_1 \in \mathbb{R}^M$, and given a sequence of external inputs $\boldsymbol{S} = \{\boldsymbol{s}_t\}$, we can recursively rewrite eq. (1) as

$$\boldsymbol{z}_T = F_{\boldsymbol{\theta}}(F_{\boldsymbol{\theta}}(F_{\boldsymbol{\theta}}(...F_{\boldsymbol{\theta}}(\boldsymbol{z}_1, \boldsymbol{s}_2)...))) =: F_{\boldsymbol{\theta}}^{T-1}(\boldsymbol{z}_1, \boldsymbol{s}_2). \tag{2}$$

In DS theory, we characterize the long-term behavior of such sequences by its spectrum of Lyapunov exponents. The Lyapunov exponents estimate the exponential growth rates in different local directions of the system's state space, and the largest Lyapunov exponent gives the dominant exponential behavior. Let us denote the system's Jacobian at time $t$ by

$$\boldsymbol{J}_t := \frac{\partial F_{\boldsymbol{\theta}}(\boldsymbol{z}_{t-1}, \boldsymbol{s}_t)}{\partial \boldsymbol{z}_{t-1}} = \frac{\partial \boldsymbol{z}_t}{\partial \boldsymbol{z}_{t-1}}. \tag{3}$$

For instance, for standard RNNs we would have $J_t = W \, diag\big(f'(W z_{t-1} + B s_t + h)\big)$, where $diag$ denotes a diagonal matrix for which the $i$-th diagonal entry is the derivative of $f$ w.r.t. $z_{i,t-1}$. Then, the maximal Lyapunov exponent along an RNN trajectory $\{z_1, z_2, \cdots, z_T, \cdots\}$ is defined as

$$\lambda_{max} := \lim_{T \to \infty} \frac{1}{T} \log \left\| \prod_{r=0}^{T-2} J_{T-r} \right\|, \tag{4}$$

where $\| \cdot \|$ denotes the spectral norm (or any subordinate norm) of a matrix. If $\lambda_{max} < 0$ nearby trajectories will ultimately converge to a fixed point or cycle, while for $\lambda_{max} > 0$ (a necessary condition for chaos) initially nearby trajectories will exponentially separate, i.e. we will have divergence along one (or more) directions in state space. This accounts for the sensitive dependence on initial conditions in chaotic systems.

Now let $\mathcal{L}(W, B, h)$ be some loss function employed for RNN training that decomposes in time as $\mathcal{L} = \sum_{t=1}^{T} \mathcal{L}_t$. Suppose we fancy BPTT as our training algorithm (similar derivations could be performed for RTRL), we recursively develop the loss gradients w.r.t. some RNN parameter $\theta$ in time (i.e., across layers of the RNN unrolled in time) as

$$\frac{\partial \mathcal{L}}{\partial \theta} = \sum_{t=1}^{T} \frac{\partial \mathcal{L}_t}{\partial \theta} \quad \text{with} \quad \frac{\partial \mathcal{L}_t}{\partial \theta} = \sum_{r=1}^{t} \frac{\partial \mathcal{L}_t}{\partial z_t} \frac{\partial z_t}{\partial z_r} \frac{\partial^+ z_r}{\partial \theta}, \tag{5}$$

and

$$\frac{\partial z_t}{\partial z_r} = \frac{\partial z_t}{\partial z_{t-1}} \frac{\partial z_{t-1}}{\partial z_{t-2}} \cdots \frac{\partial z_{r+1}}{\partial z_r} = \prod_{k=0}^{t-r-1} \frac{\partial z_{t-k}}{\partial z_{t-k-1}} = \prod_{k=0}^{t-r-1} J_{t-k}, \tag{6}$$

where $\partial^+$ denotes the immediate derivative. Now observe that the behavior of the loss gradients crucially depends on the *product series* of Jacobians in eqn. (6) : If the maximum absolute eigenvalues of the Jacobians $J_t$ will, in the geometric mean, be larger than 1 (i.e., $\left\| \prod_{r=0}^{T-2} J_{T-r} \right\|^{1/T} > 1$), gradients will explode as $T \to \infty$, while they will saturate if $\left\| \prod_{r=0}^{T-2} J_{T-r} \right\|^{1/T} < 1$. Thus, the key point to note is that the same terms that occur in the definition of the Lyapunov spectrum, eqn. (4), resurface in the loss gradients, eqn. (5) & (6). This accounts for the tight links between system dynamics and gradients.

## 3.2 Fixed points and cyclic dynamics

Let us start by considering the simplest types of dynamics that can occur in RNNs (or any discrete-time DS): fixed points and cycles. In fact, by far most of the literature on global stability in RNNs and on loss gradients focused on just fixed points (Chang et al., 2019; Kolter & Manek, 2019; Erichson et al., 2021), with only few authors who recently started to also connect cyclic behavior to loss gradients (Schmidt et al., 2021; Rusch & Mishra, 2021a). Recall that a fixed point of a recursive map $z_t = F(z_{t-1})$ is defined as a point $z^*$ for which we have $z^* = F(z^*)$.[2] Likewise, a $k$-cycle ($k > 1$) is a set of temporally consecutive periodic points $P_k := \{z_{t_1}, z_{t_2}, \ldots, z_{t_k}\} = \{z_{t_1}, F(z_{t_1}), \ldots, F^{k-1}(z_{t_1})\}$ that we obtain from recursive application of the map such that each of the cyclic points $z_{t_r} \in P_k$ is a fixed point of the $k$ times iterated map $F^k$ (with $k$ being the smallest positive integer for which this holds). To simplify the subsequent treatment, we will collectively refer to fixed points and cycles as $k$-cycles ($k \geq 1$). Further recall that a fixed point or $k$-cycle is called *stable* if the maximum absolute eigenvalue of the Jacobian evaluated at that point is smaller than 1, *neutrally stable* if exactly 1, and *unstable* otherwise. Although the results we develop in this and the following sections will hold more widely, we will restrict our attention to recursive maps $F_{\boldsymbol{\theta}}$ from the class of RNNs $\mathcal{R} = \{\texttt{standardRNN, LSTM, GRU, PLRNN}\}$ (see Appx. A.1 for details).

Based on the observations made in the previous sections we can state the following theorem that links RNN dynamics and loss gradients:

**Theorem 1.** *Consider an RNN $F_{\boldsymbol{\theta}} \in \mathcal{R}$ parameterized by $\boldsymbol{\theta}$, and assume that it converges to a stable fixed point or $k$-cycle $\Gamma_k$ ($k \geq 1$) with $\mathcal{B}_{\Gamma_k}$ as its basin of attraction[3]. Then for every $z_1 \in \mathcal{B}_{\Gamma_k}$ (i)*

---

[2]From here on we will suppress the explicit dependence on external inputs $s_t$ notation-wise, see Remark 2.

[3]The basin of attraction is defined as the set of all points from which orbits converge to the resp. attractor.

the Jacobian $\frac{\partial z_T}{\partial z_1}$ exponentially vanishes as $T \to \infty$; (ii) for $\Gamma_k$ the tangent vectors $\frac{\partial z_T}{\partial \theta}$ and thus the gradient of the loss function, $\frac{\partial \mathcal{L}_T}{\partial \theta}$, will be bounded from above, i.e. will not diverge for $T \to \infty$; and (iii) for the PLRNN (27) both $\left\| \frac{\partial z_T}{\partial \theta} \right\|$ and $\left\| \frac{\partial \mathcal{L}_T}{\partial \theta} \right\|$ will remain bounded for every $z_1 \in \mathcal{B}_{\Gamma_k}$ as $T \to \infty$.

*Proof.* $(i)$ Assume that $\Gamma_k$ is a stable $k$-cycle $(k \geq 1)$ denoted by

$$\Gamma_k = \{z_1, z_2, \cdots, z_T, \cdots\} = \{z_{t^{*k}}, z_{t^{*k}-1}, \cdots, z_{t^{*k}-(k-1)}, z_{t^{*k}}, z_{t^{*k}-1}, \cdots, z_{t^{*k}-(k-1)}, \cdots\}. \tag{7}$$

Then, the largest Lyapunov exponent of $\Gamma_k$ is given by

$$\lambda_{\Gamma_k} = \lim_{t \to \infty} \frac{1}{t} \ln \left\| J_t^* J_{t-1}^* \cdots J_2^* \right\| = \lim_{j \to \infty} \frac{1}{jk} \ln \left\| \left( \prod_{s=0}^{k-1} J_{t^{*k}-s} \right)^j \right\|. \tag{8}$$

By assumption of stability of $\Gamma_k$ we have $\lambda_{\Gamma_k} < 0$ and also $\rho\left( \prod_{s=0}^{k-1} J_{t^{*k}-s} \right) < 1$, which implies

$$\lim_{t \to \infty} J_t^* J_{t-1}^* \cdots J_2^* = \lim_{j \to \infty} \left( \prod_{s=0}^{k-1} J_{t^{*k}-s} \right)^j = 0. \tag{9}$$

Now suppose that $\mathcal{O}_{z_1}$ is an orbit of (1) converging to $\Gamma_k$, i.e. $z_1 \in \mathcal{B}_{\Gamma_k}$. Since $\mathcal{O}_{z_1}$ and $\Gamma_k$ have the same largest Lyapunov exponent, we have

$$\lambda_{\mathcal{O}_{z_1}} = \lim_{T \to \infty} \frac{1}{T} \ln \left\| J_T J_{T-1} \cdots J_2 \right\| = \lambda_{\Gamma_k} < 0, \tag{10}$$

and hence for $z_1 \in \mathcal{B}_{\Gamma_k}$

$$\lim_{T \to \infty} \left\| \frac{\partial z_T}{\partial z_1} \right\| = \lim_{T \to \infty} \left\| J_T J_{T-1} \cdots J_2 \right\| = 0. \tag{11}$$

$(ii) \& (iii)$ See Appx. A.2.1. $\qquad\qquad\qquad\qquad\qquad\qquad\qquad\qquad\qquad\qquad\qquad\qquad \square$

**Remark 1.** *The result of Theorem 1 part $(i)$ will be generally true for any first-order-Markovian recursive map (1), but the conclusions in part $(ii)$ may hinge on its specific definition.*

**Remark 2.** *None of the results above and throughout sect. 3 require the dynamics to be autonomous, the theory applies whether there is external input or not. In fact, mathematically, non-autonomous (externally forced) systems can always be rewritten as autonomous dynamical systems (Alligood et al., 1996; Perko, 2001; Zhang et al., 2009), see Appx. A.1.1 for details.*

The results above ensure that loss gradients will not diverge (explode) as $T \to \infty$ in RNNs that are "well-behaved" in the sense that they converge to a fixed point or cycle. This is a generalization of the results given in Theorem 1 in Schmidt et al. (2021), where this was shown only a) for the specific class of PLRNNs and b) for specific constraints imposed on the eigenvalue spectrum of the RNN's Jacobians which were relaxed in our theorem above.

While our treatment above is centered on the "exploding-gradients" case, various architectural modifications or regularization techniques can ensure that gradients do not vanish either, i.e. remain bounded from below as well. This was established, for instance, in Schmidt et al. (2021) for PLRNNs using 'manifold attractor regularization'. In Appx. A.2.1 we show that the results from Theorem 2 from Schmidt et al. (2021) on *doubly* bounded (from below and above) loss gradients can indeed be extended to the more general case covered by Theorem 1 above.

### 3.3 Chaotic dynamics

We will now consider the all-important chaotic case. Let $F$ be a recursive map and $\mathcal{O}_{z_1} = \{z_1, z_2, z_3, \cdots\}$ be an orbit of $F$. The orbit is chaotic if (i) it is not asymptotically periodic and (ii) has at least one positive Lyapunov exponent (Glendinning & Simpson, 2021; Meiss, 2007). If the system's invariant set is *bounded*, condition (ii) is considered a standard signature of chaos, as in this case two nearby orbits separate exponentially fast, but at the same time their mutual separation cannot go to infinity so that there are also folds. The following theorem states the sufficient condition for exploding gradients:

**Theorem 2.** *Suppose that an RNN $F_{\boldsymbol{\theta}} \in \mathcal{R}$ (parameterized by $\boldsymbol{\theta}$) has a chaotic attractor $\Gamma^*$ with $\mathcal{B}_{\Gamma^*}$ as its basin of attraction. Then, for every orbit with $\boldsymbol{z}_1 \in \mathcal{B}_{\Gamma^*}$, (i) the Jacobians connecting temporally distal states $\boldsymbol{z}_T$ and $\boldsymbol{z}_t$ ( $T \gg t$), $\frac{\partial \boldsymbol{z}_T}{\partial \boldsymbol{z}_t}$, will exponentially explode for $T \to \infty$, and (ii) the tangent vector $\frac{\partial \boldsymbol{z}_T}{\partial \theta}$ and so the gradients of the loss function, $\frac{\partial \mathcal{L}_T}{\partial \theta}$, will diverge as $T \to \infty$.*

*Proof.* Let the RNN $F_{\boldsymbol{\theta}} \in \mathcal{R}$ have a chaotic orbit denoted by $\Gamma^* = \{\boldsymbol{z}_1^*, \boldsymbol{z}_2^*, \cdots, \boldsymbol{z}_T^*, \cdots\}$. Then, denoting by $J_T^*$ the Jacobian of (1) at $\boldsymbol{z}_T^* \in \Gamma^*$, the largest Lyapunov exponent of $\Gamma^*$ is given by

$$\lambda = \lim_{T \to \infty} \frac{1}{T} \ln \left\| J_T^* J_{T-1}^* \cdots J_2^* \right\|. \tag{12}$$

Since $\Gamma^*$ is chaotic, so $\lambda > 0$. Hence, from (12), it is concluded that

$$\lim_{T \to \infty} \left\| J_T^* J_{T-1}^* \cdots J_2^* \right\| = \lim_{T \to \infty} \left\| \frac{\partial \boldsymbol{z}_T^*}{\partial \boldsymbol{z}_t^*} \right\| = \infty \ , \qquad T \gg t. \tag{13}$$

Now, according to Oseledec's multiplicative ergodic Theorem, nearly all the points in the basin of attraction of $\Gamma^*$ have the same largest Lyapunov exponent $\lambda$. Thus, (13) holds for every $\boldsymbol{z}_1 \in \mathcal{B}_{\Gamma^*}$.

$(ii)$ See Appx. A.2.2. $\qquad\qquad\qquad\qquad\qquad\qquad\qquad\qquad\qquad\qquad\qquad\qquad \square$

**Remark 3.** *The first part of Theorem 2 holds for all first-order-Markovian recursive maps (1). Note that for LSTMs, $\frac{\partial \boldsymbol{z}_T}{\partial \boldsymbol{z}_t}$ ($\boldsymbol{z} := (\boldsymbol{h}, \boldsymbol{c})^\mathsf{T}$) denotes the full Jacobian of both hidden and cell states.*

We collect some further mathematical results and remarks related to Theorem 2 in Appx. A.3.1.

Hence, the essential result is that for all popular RNNs $\mathcal{R}$ and activation functions, loss gradients will inevitably diverge if the RNN latent states converge to a chaotic attractor.

## 3.4 QUASI-PERIODICITY

Quasi-periodicity is a long-term behavior which occurs on a torus and, superficially, bears some similarity to chaos in the sense that, strictly speaking, orbits are also *aperiodic*. That is, as $T \to \infty$, trajectories will never close up with themselves. Moreover, every trajectory becomes arbitrarily close to any point on the torus, that is, it is dense. One important difference between quasi-periodic and chaotic systems is, however, that in a quasi-periodic system, as time passes, two close initial conditions are *linearly* diverging, while in a chaotic system the divergence is exponential.

**Theorem 3.** *Assume that an RNN $F_{\boldsymbol{\theta}} \in \mathcal{R}$ (parameterized by $\boldsymbol{\theta}$) has a quasi-periodic attractor $\Gamma$ with $\mathcal{B}_\Gamma$ as its basin of attraction. Then, for every orbit $\boldsymbol{z}_1 \in \mathcal{B}_\Gamma$*

$$\forall 0 < \epsilon < 1 \ \exists T_0 > 1 \ s.t. \ \forall T \geq T_0 \implies (1-\epsilon)^{T-1} < \left\| \frac{\partial \boldsymbol{z}_T}{\partial \boldsymbol{z}_1} \right\| < (1+\epsilon)^{T-1}. \tag{14}$$

*Proof.* See Appx. A.2.3. $\qquad\qquad\qquad\qquad\qquad\qquad\qquad\qquad\qquad\qquad\qquad\qquad\qquad \square$

According to Theorem 3, for every orbit converging to a quasi-periodic attractor, the Jacobians $\frac{\partial \boldsymbol{z}_T}{\partial \boldsymbol{z}_t}$ may diverge or vanish as $T \to \infty$, but this will *not* occur exponentially fast as $T \to \infty$. Thus, even for bounded non-chaotic RNNs we may sometimes stumble into the problem of diverging gradients. Although this may be a less common scenario, we point out it may occur if we train RNNs on real data from oscillatory systems with incommensurate frequencies, as for instance encountered in electronic engineering.

In Appx. A.3.2 we have collected further mathematical results on the connection between RNN dynamics and loss gradients that hold regardless of the RNN's limiting behavior.

## 4 EMPIRICAL EVALUATION

Our theoretical results imply that chaotic time series pose a principle challenge for RNN training that cannot easily be circumvented through specifically designed architectures, constraints, or regularization criteria. If the underlying DS we aim to capture is chaotic, loss gradients propagated back in time will inevitably explode. Here we will work out some implications for RNN training and potential remedies empirically.

### 4.1 Training on systems with exploding gradients by sparse teacher forcing

To illustrate the connections between theory and RNN training, we revive the old idea of TF (Williams & Zipser, 1989) as a mechanism for truncating error gradients while training. However, we would like to do this such that important information about the system dynamics does not get lost, for which Lyapunov theory offers some guidance. Specifically, we should not force the system back onto the true trajectory all or most of the time (as in "classical TF"), but should effectively "re-calibrate" it only at certain time points chosen wisely according to the system's local divergence rates. This procedure will be referred to as *sparsely forced BPTT* in the following.

Assume we want to train an RNN with hidden states $z_t \in \mathbb{R}^M$ and linear (or affine) output layer on a time-series $\{x_1, x_2, \cdots, x_T\}$ generated by a chaotic system.[4] The linear output layer $\hat{x}_t = B z_t$, $B \in \mathbb{R}^{N \times M}$, maps the RNN hidden states into the observation space. This allows us to modify the original TF procedure by constructing a control series $\{\tilde{z}_1, \tilde{z}_2, \cdots, \tilde{z}_T\}$ from the observations by "inverting" the linear output mapping [5]

$$\tilde{z}_t = (B^\mathsf{T} B)^{-1} B^\mathsf{T} x_t. \tag{15}$$

The idea is to supply this control signal only sparsely, separated by the learning interval $\tau$ between consecutive forcings. Hence, defining $\mathcal{T} = \{n\tau + 1\}_{n \in \mathbb{N}_0}$ as the set of all time points at which we force the RNN onto the 'true' values, the RNN updates can be written as

$$z_{t+1} = \begin{cases} RNN(\tilde{z}_t) & \text{if } t \in \mathcal{T} \\ RNN(z_t) & \text{else} \end{cases}. \tag{16}$$

This forcing is applied *after* calculation of the loss, such that $\mathcal{L}_t = \|x_t - B z_t\|_2^2$ irrespective of whether $t$ is in $\mathcal{T}$ or not (and of course it is applied only during training, not at test time!). Replacing hidden states $z_t$ with their teacher-forced signals $\tilde{z}_t$ simply breaks divergence between true and predicted trajectories at time points $t \in \mathcal{T}$, and also cuts off the Jacobians by breaking the temporal contingency (for details see Appx. A.7). The learning interval $\tau$ hence controls how many time steps are included in the gradient calculation and has to be chosen with care such as to balance the effects of exploding gradients vs. those of loosing relevant time scales and long-term dependencies. While it is general wisdom that an optimal batch size will facilitate training, the point here is thus much more specific: Ideally $\tau$ should be chosen in accordance with the system's Lyapunov spectrum, for instance based on the *predictability time* (Bezruchko & Smirnov, 2010)

$$\tau_{\text{pred}} = \frac{\ln 2}{\lambda_{\max}}. \tag{17}$$

We emphasize that such a simple recipe for addressing the exploding gradient problem is based on modifying the training routine, and is thus in principle applicable to any model architecture.[6]

### 4.2 Example 1: Lorenz system and externally forced Duffing oscillator in chaotic regime

Let us illustrate these ideas on two classical textbook examples of chaotic DS, the chaotic Lorenz attractor as an autonomous system, and the chaotically forced Duffing oscillator as an example with explicit external input (see Appx. A.4 for details). Trajectories were repeatedly drawn from these systems, on which we trained a PLRNN, a vanilla RNN with $\tanh$ activation function, and a LSTM by stochastic gradient descent (SGD) to minimize the MSE loss between predicted and actual observations. As optimizer we used Adam (Kingma & Ba, 2015) from PyTorch (Paszke et al., 2017) with a learning rate of $0.001$. For all models, training proceeded solely by *sparsely forced BPTT* and did not employ gradient clipping or any other technique that may interfere with optimal loss truncation.

In nonlinear DS reconstruction, we are mainly interested in reproducing *invariant* properties of the underlying system such as the attractor geometry (or topology; Takens (1981); Sauer et al. (1991)) or

---

[4]Note that in DS reconstruction one usually considers the data as observations (*unsupervised* problem); according to Remark 2, however, it is mainly a matter of the scientific question addressed whether we include certain variables as explicit inputs or as observations.

[5]To ensure invertibility, one could add a regularizer $\lambda \mathbf{I}$ to $B^\mathsf{T} B$ in eqn. (15), as in ridge regression, but we did not find this necessary in any of our examples.

[6]All code produced here is available at [placeholder].

the frequency composition (i.e., time-averaged properties), while measures like ahead-prediction errors are less meaningful especially on chaotic time series (Wood, 2010; Koppe et al., 2019). Thus, in evaluating training performance, here we follow Koppe et al. (2019) in using a Kullback-Leibler divergence $D_{stsp}$ to quantify the agreement between observed and generated probability distributions across state-space to asses the overlap in attractor geometry (Appx. A.5). Moreover, we employ a dimension-wise frequency correlation measure (PSC) to quantify the agreement of power-spectra of the observed and generated time-series (Appx. A.5).

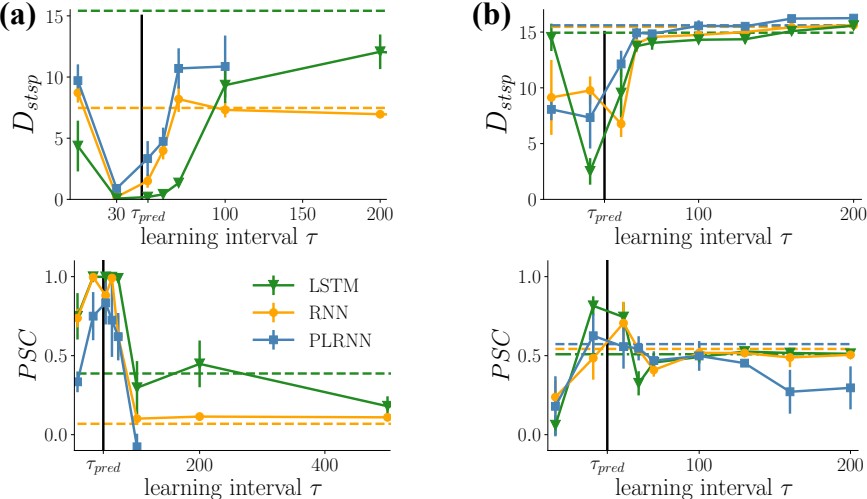

Figure 1: Overlap in attractor geometry ($D_{stsp}$, lower = better) and dimension-wise power-spectra correlations ($PSC$, higher = better) against learning interval $\tau$ for (a) the Lorenz and (b) the chaotically forced Duffing oscillator. Continuous lines = sparsely forced BPTT. Dashed lines = classical BPTT with gradient clipping. Prediction time indicated vertically in black.

Fig. 1 shows the dependence of the reconstruction quality on the learning interval $\tau$ for all RNN architectures on (a) the Lorenz and (b) the externally forced Duffing system. Fig. 2 provides particular examples of reconstructions for $\tau$ chosen too small, too large, or about right. For all models we find a system-dependent range for the optimal learning interval that agrees well with the predictability time defined in eqn. (17), where estimates for the maximal Lyapunov exponent were taken from the literature (Rosenstein et al., 1993; Gilpin, 2021). As a reference, dashed lines represent the reconstruction performance for all architectures when trained with classical BPTT and gradient clipping. The training procedure was the same as for sparsely forced BPTT, except that instead of supplying a control-signal, gradients were normalized to 1 prior to each parameter update. As evidenced by the much worse performance, gradient clipping does not effectively address the EVGP, *even for LSTMs*. As further shown in Fig. 9 in Appx. A.6.4, using the optimal window length $\tau$ but resetting the initial condition to zero (instead of its control value $\tilde{z}_t$) for each chunk equally destroys performance. This suggests that neither mere gradient normalization nor simple windowing are sufficient, but will wipe out essential information about the dynamics.

In Appx. A.6 we collect further results on the chaotic Rössler attractor (Fig. 5), high-dimensional Mackey-Glass equations (Fig. 7), and the Lorenz attractor with partial observations (Fig. 8).

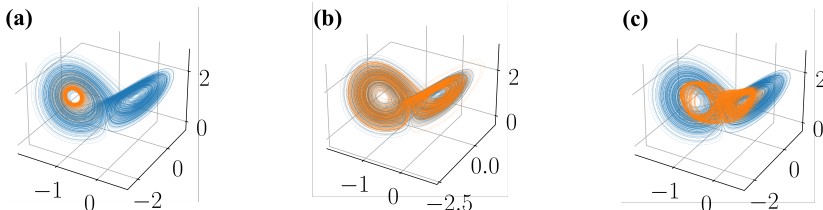

Figure 2: Lorenz attractor (blue) and example reconstructions by a LSTM (orange) trained with a learning interval (a) chosen too small ($\tau = 5$), (b) chosen optimally ($\tau = 30$), and (c) chosen too large ($\tau = 200$).

### 4.3 EXAMPLE 2: CHAOTIC WEATHER DATA

As for an empirical example, we trained all RNNs (vanilla RNN, PLRNN, LSTM) on a temperature time series recorded at the Weather Station at the Max Planck Institute for Biogeochemistry in Jena, Germany. To expose the chaotic behavior and obtain a robust estimate of the maximal Lyapunov exponent, trends and yearly cycles were removed, and nonlinear noise-reduction was performed (Kantz et al. (1993); Appx. A.4). The maximal Lyapunov exponent was determined with the *TISEAN* package (Hegger et al., 1999), as shown in Figure 3 (a). The value obtained is in close agreement with the literature (Millán et al., 2010).

Figure 3 shows that also for these empirical data the optimal training interval $\tau$ agrees well with the predictability time, eqn. (17), for all trained RNNs. Furthermore, as was the case for the DS benchmarks, gradient clipping was not able to satisfactorily tackle the EVGP, even when paired with architectures like LSTMs explicitly designed for alleviating this problem. Similar results are reported for another real-world dataset, electroencephalogram (EEG) recordings, in Appx. A.11.

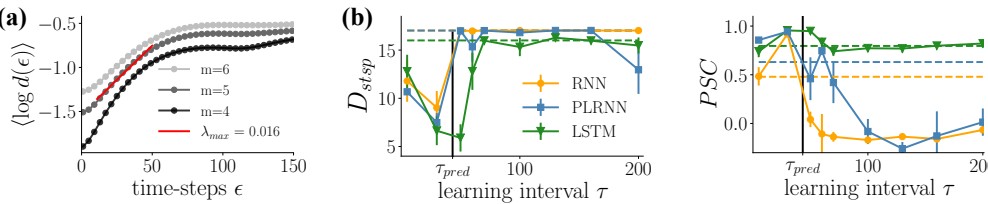

Figure 3: (a) The maximal Lyapunov exponent was determined as the slope of the average log-divergence of nearest neighbors in embedding space ($m$ = embedding dimension). (b) Reconstruction quality assessed by attractor overlap (lower = better) and power-spectrum correlation (higher = better). Black vertical lines = $\tau_{\text{pred}}$.

## 5 DISCUSSION AND CONCLUSIONS

In this paper we proved that RNN dynamics and loss gradients are intimately related for all major types of RNNs and activation functions. If the RNN is "well behaved" in the sense that its dynamics converges to a fixed point or cycle, loss gradients will remain bounded, and established remedies (Hochreiter & Schmidhuber, 1997; Schmidt et al., 2021) can be used to refrain them from vanishing. However, if the dynamics are chaotic, gradients will always explode. This constitutes a *principle* problem in RNN training that cannot easily be mastered through architectural design or gradient clipping. It is furthermore a practically highly relevant one, as most time series we encounter in nature, and many from man-made systems as well, are inherently chaotic. While we do not offer a full solution to this problem here, we suggest it might be tackled in training by taking a system's local divergence rates as measured through the Lyapunov spectrum into account. Hence, rather than conquering the EVGP by structural design or specific constraints or regularization terms, we recommend to put the focus more on the training process itself. We illustrated this point empirically using *sparsely forced BPTT*, a training technique that pulls trajectories back on track at times determined by the maximal Lyapunov exponent. Doing so leads to optimal reconstruction results for a variety of simulated and real-world benchmarks, regardless of the specific RNN architecture employed in training. We stress that our goal above all was to provide a mathematically grounded perspective on the problem, with the empirical section focused on elucidating the practical implications of the theoretical results. Empirically, for instance, precise Lyapunov exponents may sometimes be hard to obtain, although our empirical examples confirm that estimates based on log-divergence plots may work sufficiently well.

ACKNOWLEDGMENTS

ETHICS STATEMENT

The current work deals with theoretical-mathematical aspects of RNN training and performs basic research on limitations of training algorithms. As such it has no direct ethical implications. Since many real world applications depend on accurate forward predictions of time series, however, this

paper may raise awareness for an important issue that will also be relevant in practical settings, including sensitive domains like medical time series, weather forecasts, or traffic control.

## REPRODUCIBILITY STATEMENT

All theoretical results in this paper were carefully and thoroughly proven, with all proofs and detailed derivations available in the Appendix. Likewise, we will make available all code used in the empirical section in a way that will allow others to easily reproduce the results from this paper. This means we will include everything, starting with the code for benchmark simulations and simulated time series data used for evaluation, code for our model and training algorithms, up to the meta-files that produce the actual figures in this work, on our lab github site. All of this will be clearly documented.

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

## A APPENDIX

### A.1 THEOREMS: PRELIMINARIES

#### A.1.1 TRANSFORMING NON-AUTONOMOUS INTO AUTONOMOUS DISCRETE-TIME DS

Following (Zhang et al., 2009), and based on similar reasoning as for continuous time (ODE-based) DS (Alligood et al., 1996; Perko, 2001), let us consider the non-autonomous discrete-time DS

$$\boldsymbol{x}_{t+1} = F(\boldsymbol{x}_t, t), \qquad \boldsymbol{x} \in \mathbb{R}^n. \tag{18}$$

Defining $\boldsymbol{z}_t = (\boldsymbol{x}_t, t)^\mathsf{T}$ and $G(\boldsymbol{z}_t) = (F(\boldsymbol{x}_t, t), t+1)^\mathsf{T}$, system (18) can be rewritten as the autonomous system

$$\boldsymbol{z}_{t+1} = G(\boldsymbol{z}_t), \qquad \boldsymbol{z} \in \mathbb{R}^{n+1}. \tag{19}$$

Hence, in all our theoretical treatment we can confine our attention to systems of the form eqn. 18.

#### A.1.2 RNN DERIVATIVES

Considering the loss function $\mathcal{L} = \sum_{t=1}^T \mathcal{L}_t$ of an RNN $F_{\boldsymbol{\theta}} \in \mathcal{R}$ parameterized by $\boldsymbol{\theta}$, we have

$$\frac{\partial \mathcal{L}}{\partial \theta} = \sum_{t=1}^T \frac{\partial \mathcal{L}_t}{\partial \theta}, \tag{20}$$

where

$$\frac{\partial \mathcal{L}_t}{\partial \theta} = \frac{\partial \mathcal{L}_t}{\partial \boldsymbol{z}_t} \frac{\partial \boldsymbol{z}_t}{\partial \theta}. \tag{21}$$

The tangent vector $\frac{\partial \boldsymbol{z}_T}{\partial \theta}$ has the form

$$\frac{\partial \boldsymbol{z}_T}{\partial \theta} = \frac{\partial^+ \boldsymbol{z}_T}{\partial \theta} + \sum_{t=1}^{T-2} \left( \prod_{r=0}^{t-1} \boldsymbol{J}_{T-r} \right) \frac{\partial^+ \boldsymbol{z}_{T-t}}{\partial \theta}, \tag{22}$$

where $\partial^+$ denotes the immediate partial derivative. Since for an RNN $F_{\boldsymbol{\theta}} \in \mathcal{R}$ the activation function is element-wise, with $\theta$ the $m$-th element of a parameter vector $\boldsymbol{\theta}$ (or belonging to the $m$-th row of a parameter matrix $\boldsymbol{\theta}$), we have

$$\frac{\partial^+ \boldsymbol{z}_T}{\partial \theta} = \begin{pmatrix} 0 & \cdots & 0 & \frac{\partial^+ z_{m,T}}{\partial \theta} & 0 & \cdots & 0 \end{pmatrix}^\mathsf{T}. \tag{23}$$

For instance, let $\boldsymbol{\theta} = \boldsymbol{W}$ be a weight matrix, then

$$\frac{\partial \mathcal{L}}{\partial \boldsymbol{W}} = \begin{pmatrix} \frac{\partial \mathcal{L}}{\partial w_{11}} & \frac{\partial \mathcal{L}}{\partial w_{12}} & \cdots & \frac{\partial \mathcal{L}}{\partial w_{1M}} \\ \frac{\partial \mathcal{L}}{\partial w_{21}} & \frac{\partial \mathcal{L}}{\partial w_{22}} & \cdots & \frac{\partial \mathcal{L}}{\partial w_{2M}} \\ \vdots & & & \\ \frac{\partial \mathcal{L}}{\partial w_{M1}} & \frac{\partial \mathcal{L}}{\partial w_{M2}} & \cdots & \frac{\partial \mathcal{L}}{\partial w_{MM}} \end{pmatrix}. \tag{24}$$

In this case, for the standard RNN we have

$$\frac{\partial^+ \boldsymbol{z}_T}{\partial w_{mk}} = \begin{pmatrix} 0 & \cdots & 0 & z_{k,T-1}\, \xi_{mk}(\boldsymbol{z}_{T-1}) & 0 & \cdots & 0 \end{pmatrix}^\mathsf{T} = \mathbf{1}_{(m,k)}\, \xi_{mk}(\boldsymbol{z}_{T-1})\, \boldsymbol{z}_{T-1}, \tag{25}$$

where $\xi_{mk}(\boldsymbol{z}_{T-1}) = f'_{w_{m,k}}\left( \sum_{j=1}^M w_{mj}\, z_{j,T-1} + \sum_{j=1}^M b_{mj}\, s_{j,T} + h_m \right)$, and $f'_{w_{m,k}}$ stands for the derivative of $f$ with respect to $w_{m,k}$.

Therefore, for standard RNNs, (22) becomes

$$\frac{\partial \boldsymbol{z}_T}{\partial w_{mk}} = \mathbf{1}_{(m,k)}\, \xi_{mk}(\boldsymbol{z}_{T-1})\, \boldsymbol{z}_{T-1} + \sum_{t=1}^{T-2} \left( \prod_{r=0}^{t-1} \boldsymbol{J}_{T-r} \right) \mathbf{1}_{(m,k)}\, \xi_{mk}(\boldsymbol{z}_{T-t-1})\, \boldsymbol{z}_{T-t-1}. \tag{26}$$

### A.1.3 PIECEWISE-LINEAR RNN (PLRNN)

The PLRNN has the generic form (Koppe et al., 2019; Schmidt et al., 2021)

$$\boldsymbol{z}_t = F(\boldsymbol{z}_{t-1}) = \boldsymbol{A}\,\boldsymbol{z}_{t-1} + \boldsymbol{W}\phi(\boldsymbol{z}_{t-1}) + \boldsymbol{C}\boldsymbol{s}_t + \boldsymbol{h} + \boldsymbol{\varepsilon}_t, \tag{27}$$

where $\phi(\boldsymbol{z}_{t-1}) = \max(\boldsymbol{z}_{t-1}, 0)$ is the element-wise rectified linear unit (ReLU) function, $\boldsymbol{z}_t \in \mathbb{R}^M$ is the neural state vector, $\boldsymbol{A} \in \mathbb{R}^{M \times M}$ is a diagonal matrix of auto-regression weights, $\boldsymbol{W} \in \mathbb{R}^{M \times M}$ is a matrix of connection weights, $\boldsymbol{h} \in \mathbb{R}^M$ is the bias vector, $\boldsymbol{s}_t \in \mathbb{R}^K$ the external input weighted by $\boldsymbol{C} \in \mathbb{R}^{M \times K}$, and $\boldsymbol{\varepsilon}_t \sim \mathcal{N}(0, \boldsymbol{\Sigma})$ a Gaussian noise term with diagonal covariance matrix $\boldsymbol{\Sigma}$.

Equation (27) can be rewritten as

$$\boldsymbol{z}_t = (\boldsymbol{A} + \boldsymbol{W}\boldsymbol{D}_{\Omega(t-1)})\boldsymbol{z}_{t-1} + \boldsymbol{C}\boldsymbol{s}_t + \boldsymbol{h} + \boldsymbol{\varepsilon}_t =: \boldsymbol{W}_{\Omega(t-1)}\,\boldsymbol{z}_{t-1} + \boldsymbol{C}\boldsymbol{s}_t + \boldsymbol{h} + \boldsymbol{\varepsilon}_t, \tag{28}$$

where $\boldsymbol{D}_{\Omega(t)} := \mathrm{diag}(\boldsymbol{d}_{\Omega(t)})$ is a diagonal matrix with $\boldsymbol{d}_{\Omega(t)} := (d_1, d_2, \cdots, d_M)$ an indicator vector such that $d_m(z_{m,t}) =: d_m = 1$ whenever $z_{m,t} > 0$, and zeros otherwise.

For the PLRNN (28) we have

$$\boldsymbol{J}_t = \frac{\partial \boldsymbol{z}_t}{\partial \boldsymbol{z}_{t-1}} = \boldsymbol{W}_{\Omega(t-1)}, \tag{29}$$

and $\left\| \boldsymbol{W}_{\Omega(t-1)} \right\| \leq \|\boldsymbol{A}\| + \|\boldsymbol{W}\|$.

Furthermore, the derivatives (22) for the PLRNN (28) are

$$\frac{\partial \boldsymbol{z}_T}{\partial w_{mk}} = \boldsymbol{1}_{(m,k)}\boldsymbol{D}_{\Omega(T-1)}\,\boldsymbol{z}_{T-1} + \sum_{j=2}^{T-1} \left( \prod_{i=1}^{j-1} \boldsymbol{W}_{\Omega(T-i)} \right) \boldsymbol{1}_{(m,k)}\boldsymbol{D}_{\Omega(T-j)}\,\boldsymbol{z}_{T-j}. \tag{30}$$

### A.1.4 LONG SHORT-TERM MEMORY (LSTM)

The LSTM is defined by the equations

$$\begin{aligned}
\boldsymbol{i}_t &= \sigma\big(\boldsymbol{W}_{ii}\boldsymbol{s}_t + \boldsymbol{W}_{hi}\boldsymbol{h}_{t-1} + \boldsymbol{b}_i\big) \\
\boldsymbol{f}_t &= \sigma\big(\boldsymbol{W}_{if}\boldsymbol{s}_t + \boldsymbol{W}_{hf}\boldsymbol{h}_{t-1} + \boldsymbol{b}_f\big) \\
\boldsymbol{g}_t &= \tanh\big(\boldsymbol{W}_{ig}\boldsymbol{s}_t + \boldsymbol{W}_{hg}\boldsymbol{h}_{t-1} + \boldsymbol{b}_g\big) \\
\boldsymbol{o}_t &= \sigma\big(\boldsymbol{W}_{io}\boldsymbol{s}_t + \boldsymbol{W}_{ho}\boldsymbol{h}_{t-1} + \boldsymbol{b}_o\big) \\
\boldsymbol{c}_t &= \boldsymbol{f}_t \odot \boldsymbol{c}_{t-1} + \boldsymbol{i}_t \odot \boldsymbol{g}_t \\
\boldsymbol{h}_t &= \boldsymbol{o}_t \odot \tanh\left(\boldsymbol{c}_t\right)
\end{aligned} \tag{31}$$

where $\{\boldsymbol{s}_t\}$ is the input sequence, $\boldsymbol{W}$ denotes weight matrices, $\boldsymbol{b}$ bias terms, $\boldsymbol{i}_t, \boldsymbol{f}_t, \boldsymbol{g}_t, \boldsymbol{o}_t$ demonstrate the input, forget, cell, and output gates, $\boldsymbol{h}_t$ and $\boldsymbol{c}_t$ are the hidden and cell states at time $t$ respectively, $\sigma$ is the sigmoid activation function, and $\odot$ represents the element-wise (Hadamard) product (see (Hochreiter & Schmidhuber, 1997; Graves et al., 2016; Vlachas et al., 2018) for further information on LSTMs).

Defining $\boldsymbol{z}_t := (\boldsymbol{h}_t, \boldsymbol{c}_t)^{\mathsf{T}}$, the LSTM (31) can be represented as the first-order recursive map

$$\boldsymbol{z}_t = F_{\boldsymbol{\theta}}(\boldsymbol{z}_{t-1}) = \begin{pmatrix} \boldsymbol{o}_t \odot \tanh\left(\boldsymbol{f}_t \odot \boldsymbol{c}_{t-1} + \boldsymbol{i}_t \odot \boldsymbol{g}_t\right) \\ \boldsymbol{f}_t \odot \boldsymbol{c}_{t-1} + \boldsymbol{i}_t \odot \boldsymbol{g}_t \end{pmatrix}. \tag{32}$$

The term $\frac{\partial \mathcal{L}_t}{\partial \theta}$ in (20) for some LSTM parameter $\theta$ can be written as

$$\frac{\partial \mathcal{L}_t}{\partial \theta} = \sum_{r=1}^{t} \frac{\partial \mathcal{L}_t}{\partial \boldsymbol{h}_t} \frac{\partial \boldsymbol{h}_t}{\partial \boldsymbol{z}_t} \frac{\partial \mathcal{L}_t}{\partial \boldsymbol{z}_t} \frac{\partial \boldsymbol{z}_t}{\partial \boldsymbol{z}_r} \frac{\partial \boldsymbol{z}_r}{\partial \theta}. \tag{33}$$

A necessary condition for LSTMs to have a chaotic orbit is given by:

**Proposition 1.** *Let the LSTM given by (31) have a chaotic attractor $\Gamma^*$ with $\mathcal{B}_{\Gamma^*}$ as its basin of attraction. Then for every $\boldsymbol{z}_1 = (\boldsymbol{h}_1, \boldsymbol{c}_1)^\mathsf{T} \in \mathcal{B}_{\Gamma^*}$*

$$\gamma := \lim_{T \to \infty} \sqrt[T]{\left\| \begin{pmatrix} \frac{\partial \boldsymbol{h}_T}{\partial \boldsymbol{h}_1} & \frac{\partial \boldsymbol{h}_T}{\partial \boldsymbol{c}_1} \\ \frac{\partial \boldsymbol{c}_T}{\partial \boldsymbol{h}_1} & \frac{\partial \boldsymbol{c}_T}{\partial \boldsymbol{c}_1} \end{pmatrix} \right\|} > 1. \tag{34}$$

*Proof.* The Jacobian matrix of (32) for $t > 1$ can be written in the block form

$$\frac{\partial \boldsymbol{z}_t}{\partial \boldsymbol{z}_{t-1}} = J_t = \begin{pmatrix} \frac{\partial \boldsymbol{h}_t}{\partial \boldsymbol{h}_{t-1}} & \frac{\partial \boldsymbol{h}_t}{\partial \boldsymbol{c}_{t-1}} \\ \frac{\partial \boldsymbol{c}_t}{\partial \boldsymbol{h}_{t-1}} & \frac{\partial \boldsymbol{c}_t}{\partial \boldsymbol{c}_{t-1}} \end{pmatrix}. \tag{35}$$

Further, due to the chain rule, we have

$$\begin{aligned}
J_t \, J_{t-1} &= \begin{pmatrix} \frac{\partial \boldsymbol{h}_t}{\partial \boldsymbol{h}_{t-1}} \frac{\partial \boldsymbol{h}_{t-1}}{\partial \boldsymbol{h}_{t-2}} + \frac{\partial \boldsymbol{h}_t}{\partial \boldsymbol{c}_{t-1}} \frac{\partial \boldsymbol{c}_{t-1}}{\partial \boldsymbol{h}_{t-2}} & \frac{\partial \boldsymbol{h}_t}{\partial \boldsymbol{h}_{t-1}} \frac{\partial \boldsymbol{h}_{t-1}}{\partial \boldsymbol{c}_{t-2}} + \frac{\partial \boldsymbol{h}_t}{\partial \boldsymbol{c}_{t-1}} \frac{\partial \boldsymbol{c}_{t-1}}{\partial \boldsymbol{c}_{t-2}} \\ \frac{\partial \boldsymbol{c}_t}{\partial \boldsymbol{h}_{t-1}} \frac{\partial \boldsymbol{h}_{t-1}}{\partial \boldsymbol{h}_{t-2}} + \frac{\partial \boldsymbol{c}_t}{\partial \boldsymbol{c}_{t-1}} \frac{\partial \boldsymbol{c}_{t-1}}{\partial \boldsymbol{h}_{t-2}} & \frac{\partial \boldsymbol{c}_t}{\partial \boldsymbol{h}_{t-1}} \frac{\partial \boldsymbol{h}_{t-1}}{\partial \boldsymbol{c}_{t-2}} + \frac{\partial \boldsymbol{c}_t}{\partial \boldsymbol{c}_{t-1}} \frac{\partial \boldsymbol{c}_{t-1}}{\partial \boldsymbol{c}_{t-2}} \end{pmatrix} \\
&= \begin{pmatrix} \frac{\partial \boldsymbol{h}_t}{\partial \boldsymbol{h}_{t-2}} & \frac{\partial \boldsymbol{h}_t}{\partial \boldsymbol{c}_{t-2}} \\ \frac{\partial \boldsymbol{c}_t}{\partial \boldsymbol{h}_{t-2}} & \frac{\partial \boldsymbol{c}_t}{\partial \boldsymbol{c}_{t-2}} \end{pmatrix},
\end{aligned} \tag{36}$$

and by induction we obtain

$$\frac{\partial \boldsymbol{z}_t}{\partial \boldsymbol{z}_1} = J_t \, J_{t-1} \, J_{t-2} \cdots J_2 = \begin{pmatrix} \frac{\partial \boldsymbol{h}_t}{\partial \boldsymbol{h}_1} & \frac{\partial \boldsymbol{h}_t}{\partial \boldsymbol{c}_1} \\ \frac{\partial \boldsymbol{c}_t}{\partial \boldsymbol{h}_1} & \frac{\partial \boldsymbol{c}_t}{\partial \boldsymbol{c}_1} \end{pmatrix}. \tag{37}$$

Now assume that (32) has a chaotic orbit given by

$$\Gamma^* = \{\boldsymbol{z}_1^*, \boldsymbol{z}_2^*, \cdots, \boldsymbol{z}_T^*, \cdots\}. \tag{38}$$

According to (37), the largest Lyapunov exponent of $\Gamma^*$ is given by

$$\lambda_{\Gamma^*} = \lim_{T \to \infty} \frac{1}{T} \ln \left\| J_T^* J_{T-1}^* \cdots J_2^* \right\| = \lim_{T \to \infty} \frac{1}{T} \ln \left\| \begin{pmatrix} \frac{\partial \boldsymbol{h}_T^*}{\partial \boldsymbol{h}_1^*} & \frac{\partial \boldsymbol{h}_T^*}{\partial \boldsymbol{c}_1^*} \\ \frac{\partial \boldsymbol{c}_T^*}{\partial \boldsymbol{h}_1^*} & \frac{\partial \boldsymbol{c}_T^*}{\partial \boldsymbol{c}_1^*,} \end{pmatrix} \right\|.$$

Since $\Gamma^*$ is chaotic, so $\lambda_{\Gamma^*} > 0$, which gives

$$\lim_{T \to \infty} \sqrt[T]{\left\| \begin{pmatrix} \frac{\partial \boldsymbol{h}_T^*}{\partial \boldsymbol{h}_1^*} & \frac{\partial \boldsymbol{h}_T^*}{\partial \boldsymbol{c}_1^*} \\ \frac{\partial \boldsymbol{c}_T^*}{\partial \boldsymbol{h}_1^*} & \frac{\partial \boldsymbol{c}_T^*}{\partial \boldsymbol{c}_1^*} \end{pmatrix} \right\|} > 1. \tag{39}$$

Based on Oseledec's multiplicative ergodic Theorem, (39) holds for every $\boldsymbol{z}_1 \in \mathcal{B}_{\Gamma^*}$. This completes the proof. $\qquad \square$

### A.1.5 GATED RECURRENT UNIT (GRU)

A GRU network is defined by the equations

$$
\begin{aligned}
\boldsymbol{z}_t &= \sigma\big(\boldsymbol{W}_z\,\boldsymbol{s}_t \,+\, \boldsymbol{U}_z\boldsymbol{h}_{t-1} \,+\, \boldsymbol{b}_z\big) \\
\boldsymbol{r}_t &= \sigma\big(\boldsymbol{W}_r\,\boldsymbol{s}_t \,+\, \boldsymbol{U}_r\boldsymbol{h}_{t-1} \,+\, \boldsymbol{b}_r\big) \\
\boldsymbol{h}_t &= (1-\boldsymbol{z}_t)\odot\tanh\big(\boldsymbol{W}_h\,\boldsymbol{s}_t \,+\, \boldsymbol{U}_h(\boldsymbol{r}_t\odot\boldsymbol{h}_{t-1}) \,+\, \boldsymbol{b}_h\big) \,+\, \boldsymbol{z}_t\odot\boldsymbol{h}_{t-1},
\end{aligned}
\tag{40}
$$

where $\boldsymbol{r}_t$ represents the reset gate, $\boldsymbol{z}_t$ the update gate, $\boldsymbol{s}_t$ and $\boldsymbol{h}_t$ denote the inputs and the hidden state respectively, $\boldsymbol{W}_z, \boldsymbol{W}_r, \boldsymbol{W}_h \in \mathbb{R}^{M\times N}$ and $\boldsymbol{U}_z, \boldsymbol{U}_r, \boldsymbol{U}_h \in \mathbb{R}^{M\times M}$ are weight matrices, $\boldsymbol{b}_z, \boldsymbol{b}_r, \boldsymbol{b}_h \in \mathbb{R}^{M}$ are bias vectors, and $\sigma$ is the element-wise logistic sigmoid function (for more details about GRUs see Cho et al. (2014a)).

### A.1.6 UNITARY EVOLUTION RNN (URNN)

The uRNN, proposed in (Arjovsky et al., 2016), is defined as the nonlinear DS

$$
\boldsymbol{z}_t = \sigma_{\boldsymbol{b}}\big(\boldsymbol{W}\boldsymbol{z}_{t-1} + \boldsymbol{V}\boldsymbol{s}_t\big),
\tag{41}
$$

for which $\boldsymbol{W} \in U(M)$ is an unitary matrix, $\boldsymbol{V} \in \mathbb{C}^{M\times N}$, $\boldsymbol{b} \in \mathbb{R}^{M}$ is the bias parameter, $\boldsymbol{s}_t$ is the real- or complex-valued input of dimension $N$, and

$$
[\sigma_{\boldsymbol{b}}(\boldsymbol{z})]_i = [\sigma_{\text{modReLU}}(\boldsymbol{z})]_i = \begin{cases} \big(|z_i| + b_i\big)\frac{z_i}{|z_i|} & \text{if } |z_i| + b_i \geq 0 \\ 0 & \text{if } |z_i| + b_i < 0 \end{cases}.
\tag{42}
$$

**Proposition 2.** *The uRNN given by (41) cannot have any chaotic orbit.*

*Proof.* For any arbitrary orbit $\mathcal{O}_{\boldsymbol{z}_1}$ of (41) we have

$$
\|J_T\, J_{T-1} \cdots J_2\| = \left\|\prod_{k=0}^{T-2} \boldsymbol{D}_{T-k}\,\boldsymbol{W}^{\mathsf{T}}\right\|,
\tag{43}
$$

where $\boldsymbol{D}_t = diag\Big(\sigma_{\boldsymbol{b}}'\big(\boldsymbol{W}\boldsymbol{z}_{t-1} + \boldsymbol{V}\boldsymbol{s}_t\big)\Big)$. Since $\boldsymbol{W}$ is unitary and so a norm preserving matrix, it is concluded that

$$
\left\|\prod_{k=0}^{T-2} \boldsymbol{D}_{T-k}\,\boldsymbol{W}^{\mathsf{T}}\right\| \leq \prod_{k=0}^{T-2}\left\|\boldsymbol{D}_{T-k}\,\boldsymbol{W}^{\mathsf{T}}\right\| = \prod_{k=0}^{T-2}\left\|\boldsymbol{D}_{T-k}\right\| = 1,
\tag{44}
$$

which implies

$$
\lambda_{max} = \lim_{T\to\infty}\frac{1}{T}\ln\|J_T\, J_{T-1}\cdots J_2\| \leq 0.
\tag{45}
$$

This rules out the existence of chaos (since $\lambda_{max} > 0$ is a necessary condition for $\mathcal{O}_{\boldsymbol{z}_1}$ to be chaotic). $\qquad\square$

Note that, more generally, any RNN which is constrained such as to exhibit global convergence to a fixed point or cycle, by definition must have a maximum Lyapunov exponent $\lambda_{max} \leq 0$ (in accordance with Theorem 1), hence cannot exhibit chaotic behavior by definition.

## A.2 Theorems: Proofs

### A.2.1 Proof of theorem 1, parts (ii) & (iii)

*Proof.* $(ii)$ If $\boldsymbol{J}$ is the Jordan normal form of $\prod_{s=0}^{k-1} J_{t*k-s}$, then $\prod_{s=0}^{k-1} J_{t*k-s} = \boldsymbol{P}\boldsymbol{J}\boldsymbol{P}^{-1}$, where

$$\boldsymbol{J} = \begin{pmatrix} \boldsymbol{J}_{m_1}(\lambda_1) & 0 & 0 & \cdots & 0 \\ 0 & \boldsymbol{J}_{m_2}(\lambda_2) & 0 & \cdots & 0 \\ \vdots & \cdots & \ddots & \cdots & \vdots \\ 0 & \cdots & 0 & \boldsymbol{J}_{m_{p-1}}(\lambda_{p-1}) & 0 \\ 0 & \cdots & \cdots & 0 & \boldsymbol{J}_{m_p}(\lambda_p) \end{pmatrix}, \tag{46}$$

and $m_i$ is the algebraic multiplicity of each eigenvalue $\lambda_i$. Since $\rho\big(\prod_{s=0}^{k-1} J_{t*k-s}\big) < 1$, so the eigenvalue $\lambda_i$ associated with each Jordan block satisfies $|\lambda_i| < 1$ $(i = 1, \cdots, p)$. Moreover, every $m_i \times m_i$ Jordan block has the form

$$\boldsymbol{J}_{m_i}(\lambda_i) = \begin{pmatrix} \lambda_i & 1 & 0 & \cdots & 0 \\ 0 & \lambda_i & 1 & \cdots & 0 \\ \vdots & \vdots & \ddots & \ddots & \vdots \\ 0 & 0 & \cdots & \lambda_i & 1 \\ 0 & 0 & \cdots & 0 & \lambda_i \end{pmatrix}. \tag{47}$$

Accordingly

$$\left\| \left( \prod_{s=0}^{k-1} J_{t*k-s} \right)^j \right\| = \left\| \boldsymbol{P}\boldsymbol{J}^j\boldsymbol{P}^{-1} \right\| \leq p \left\| \boldsymbol{J}^j \right\|, \tag{48}$$

in which $p = \|\boldsymbol{P}\| \|\boldsymbol{P}^{-1}\|$. Furthermore, for $j \in \mathbb{N}$, $\boldsymbol{J}^j$ is a block diagonal matrix of the form

$$\boldsymbol{J}^j = \begin{pmatrix} \boldsymbol{J}_{m_1}^j(\lambda_1) & 0 & 0 & \cdots & 0 \\ 0 & \boldsymbol{J}_{m_2}^j(\lambda_2) & 0 & \cdots & 0 \\ \vdots & \cdots & \ddots & \cdots & \vdots \\ 0 & \cdots & 0 & \boldsymbol{J}_{m_{p-1}}^j(\lambda_{p-1}) & 0 \\ 0 & \cdots & \cdots & 0 & \boldsymbol{J}_{m_p}^j(\lambda_p) \end{pmatrix}, \tag{49}$$

in which every $m_i \times m_i$ Jordan block has the form

$$\boldsymbol{J}_{m_i}^j(\lambda_i) = \begin{pmatrix} \lambda_i^j & \binom{j}{1}\lambda_i^{j-1} & \binom{j}{2}\lambda_i^{j-2} & \cdots & \binom{j}{m_i-1}\lambda_i^{j-m_i+1} \\ 0 & \lambda_i^j & \binom{j}{1}\lambda_i^{j-1} & \cdots & \binom{j}{m_i-2}\lambda_i^{j-m_i+2} \\ \vdots & \vdots & \ddots & \ddots & \vdots \\ 0 & 0 & \cdots & \lambda_i^j & \binom{j}{1}\lambda_i^{j-1} \\ 0 & 0 & \cdots & 0 & \lambda_i^j \end{pmatrix}. \tag{50}$$

In addition, for every block $\boldsymbol{J}_{m_i}^j(\lambda_i)$, we have

$$\left\| \boldsymbol{J}_{m_i}^j(\lambda_i) \right\| \leq \sqrt{m_i} \left\| \boldsymbol{J}_{m_i}^j(\lambda_i) \right\|_\infty = \sqrt{m_i} \sum_{q=1}^{m_i} \left| \left( \boldsymbol{J}_{m_i}^j(\lambda_i) \right)_{1q} \right|$$

$$= \sqrt{m_i} \sum_{q=1}^{m_i} \binom{j}{q-1} |\lambda_i|^{j-q+1} = |\lambda_i|^j \sqrt{m_i} \left( |\lambda_i|^{1-m_i} \sum_{q=1}^{m_i} \binom{j}{q-1} |\lambda_i|^{m_i-q} \right)$$

$$\leq |\lambda_i|^j \, j^{m_i} \sqrt{m_i} \left( |\lambda_i|^{1-m_i} \sum_{q=1}^{m_i} |\lambda_i|^{m_i-q} \right) =: |\lambda_i|^j \, j^{m_i} \, N_{\lambda_i}. \tag{51}$$

Moreover, for any $1 < \tilde{r}_i < \frac{1}{|\lambda_i|}$, there exists some $l_i$ such that $j^{m_i} < \tilde{r}_i^j$ for $j \geq l_i$. This means for $j \geq l_i$

$$\left\| \boldsymbol{J}_{m_i}^j(\lambda_i) \right\| \leq N_{\lambda_i} \, |\tilde{r}_i \, \lambda_i|^j, \tag{52}$$

such that $|\tilde{r}_i \, \lambda_i| = \tilde{r}_i |\lambda_i| < 1$.

Besides, for $\boldsymbol{J}^j = \boldsymbol{J}_{m_1}^j(\lambda_1) \oplus \boldsymbol{J}_{m_2}^j(\lambda_2) \oplus \cdots \oplus \boldsymbol{J}_{m_p}^j(\lambda_p)$

$$\left\| \boldsymbol{J}^j \right\| = \max_{1 \leq i \leq p} \left\| \boldsymbol{J}_{m_i}^j(\lambda_i) \right\| =: \left\| \boldsymbol{J}_m^j(\lambda) \right\|. \tag{53}$$

Hence, from (48), (52) and (53), it is deduced that for $j \geq l$

$$\left\| \left( \prod_{s=0}^{k-1} J_{t^{*k}-s} \right)^j \right\| \leq p \, N_\lambda \, |\tilde{r} \, \lambda|^j =: \bar{p} \, r^j, \tag{54}$$

in which $r = |\tilde{r} \, \lambda| < 1$.

Furthermore, let for $\Gamma_k$

$$\max_{T \geq 1} \left\{ \left\| \boldsymbol{J}_T^* \right\| \right\} = \max_{0 \leq s \leq k-1} \left\{ \left\| J_{t^{*k}-s} \right\| \right\} = \bar{m},$$

$$\max_{T \geq 1} \left\{ \left\| \frac{\partial^+ \boldsymbol{z}_T}{\partial \theta} \right\| \right\} = \max_{0 \leq s \leq k-1} \left\{ \left\| \frac{\partial^+ \boldsymbol{z}_{t^{*k}-s}}{\partial \theta} \right\| \right\} = \xi,$$

$$\max_{T \geq 1} \left\{ \left\| \boldsymbol{z}_T \right\| \right\} = \max_{0 \leq s \leq k-1} \left\{ \left\| \boldsymbol{z}_{t^{*k}-s} \right\| \right\} = \bar{q}. \tag{55}$$

Hence, defining $\boldsymbol{z}_0 = 0$, for this $k$-cycle

$$\left\| \frac{\partial \boldsymbol{z}_T}{\partial \theta} \right\| = \left\| \frac{\partial^+ \boldsymbol{z}_T}{\partial \theta} + \sum_{t=1}^{T-2} \left( \prod_{r=0}^{t-1} \boldsymbol{J}_{T-r}^* \right) \frac{\partial^+ \boldsymbol{z}_{T-t}}{\partial \theta} \right\|$$

$$= \left\| \frac{\partial^+ \boldsymbol{z}_T}{\partial \theta} + \sum_{t=1}^{T-1} \left( \prod_{r=0}^{t-1} \boldsymbol{J}_{T-r}^* \right) \frac{\partial^+ \boldsymbol{z}_{T-t}}{\partial \theta} \right\|$$

$$\leq \bar{q} \xi \left( 1 + \sum_{t=1}^{T-1} \left\| \prod_{r=0}^{t-1} \boldsymbol{J}_{T-r}^* \right\| \right). \tag{56}$$

On the other hand, for $T = kj$, from (54) and (55) we have

$$\sum_{t=1}^{T-1} \left\| \prod_{r=0}^{t-1} \boldsymbol{J}_{T-r}^* \right\| = \sum_{t=1}^{kj-1} \left\| \prod_{r=0}^{t-1} \boldsymbol{J}_{kj-r}^* \right\| = \sum_{t=1}^{k-1} \left\| \prod_{r=0}^{t-1} \boldsymbol{J}_{kj-r}^* \right\| + \sum_{t=k}^{2k-1} \left\| \prod_{r=0}^{t-1} \boldsymbol{J}_{kj-r}^* \right\|$$

$$+ \sum_{t=2k}^{3k-1} \left\| \prod_{r=0}^{t-1} \boldsymbol{J}_{kj-r}^* \right\| + \cdots + \sum_{t=(j-2)k}^{(j-1)k-1} \left\| \prod_{r=0}^{t-1} \boldsymbol{J}_{kj-r}^* \right\| + \sum_{t=(j-1)k}^{kj-1} \left\| \prod_{r=0}^{t-1} \boldsymbol{J}_{kj-r}^* \right\|$$

$$= \sum_{t=1}^{k-1} \left\| \prod_{r=0}^{t-1} \boldsymbol{J}_{kj-r}^* \right\| + \sum_{i=2}^{j} \sum_{t=(i-1)k}^{ik-1} \left\| \prod_{r=0}^{t-1} \boldsymbol{J}_{kj-r}^* \right\|$$

$$\leq \left( \bar{m} + \bar{m}^2 + \cdots + \bar{m}^{k-1} \right) + \sum_{i=2}^{j} \bar{p} \left( 1 + \bar{m} + \bar{m}^2 + \cdots + \bar{m}^{k-1} \right) r^{i-1}. \tag{57}$$

Thus, considering $\left(\bar{m} + \bar{m}^2 + \cdots + \bar{m}^{k-1}\right) = \mathcal{M}$, it is deduced that

$$\lim_{T \to \infty} \left\| \frac{\partial z_T}{\partial \theta} \right\| = \lim_{j \to \infty} \left\| \frac{\partial z_{kj}}{\partial \theta} \right\| \leq \bar{q}\xi\left(1 + \mathcal{M} + \frac{\bar{p}\,r(1 + \mathcal{M})}{1 - r}\right) = \bar{\mathcal{M}} < \infty, \qquad (58)$$

which, by (21), implies $\frac{\partial \mathcal{L}_T}{\partial \theta}$ will be bounded for $T \to \infty$.

$(iii)$ Consider the PLRNN given by (27), where for simplicity we ignore the external inputs and noise terms. Let $\{z_{t_1}, z_{t_2}, z_{t_3}, \ldots\}$ be an orbit which converges to $\Gamma_k$. Hence

$$\lim_{n \to \infty} d(z_{t_n}, \Gamma_k) = 0, \qquad (59)$$

which implies there exists a neighborhood $U$ of $\Gamma_k$ and $k$ sub-sequences $\{z_{t_{km}}\}_{m=1}^{\infty}, \{z_{t_{km+1}}\}_{m=1}^{\infty}$, $\cdots, \{z_{t_{km+(k-1)}}\}_{m=1}^{\infty}$ of the sequence $\{z_{t_n}\}_{n=1}^{\infty}$ such that all these sub-sequences belong to $U$ and

a) $z_{t_{km+s}} = F^k(z_{t_{k(m-1)+s}}), s = 0, 1, 2, \cdots, k-1,$

b) $\lim_{m \to \infty} z_{t_{km+s}} = z_{t^{*k}-s}, s = 0, 1, 2, \cdots, k-1,$

c) for every $z_{t_n} \in U$ there is some $s \in \{0, 1, 2, \cdots, k-1\}$ such that $z_{t_n} \in \{z_{t_{km+s}}\}_{m=1}^{\infty}$.

In this case, for every $z_{t_n} \in U$ with $z_{t_n} \in \{z_{t_{km+s}}\}_{m=1}^{\infty}$, there exists some $\tilde{n} \in \mathbb{N}$ such that $z_{t_n} = z_{t_{k\tilde{n}+s}}$ and $\lim_{\tilde{n} \to \infty} z_{t_{k\tilde{n}+s}} = z_{t^{*k}-s}$. Therefore, continuity of $F$ results in

$$\lim_{\tilde{n} \to \infty} F(z_{t_{k\tilde{n}+s}}) = F(z_{t^{*k}-s}), \qquad (60)$$

and so by (28)

$$\lim_{\tilde{n} \to \infty} \left(W_{\Omega(t_{k\tilde{n}+s})}\, z_{t_{k\tilde{n}+s}} + h\right) = W_{\Omega(t^{*k}-s)}\, z_{t^{*k}-s} + h, \qquad (61)$$

which implies

$$\lim_{\tilde{n} \to \infty} W_{\Omega(t_{k\tilde{n}+s})}\, z_{t_{k\tilde{n}+s}} = W_{\Omega(t^{*k}-s)}\, z_{t^{*k}-s}. \qquad (62)$$

Assuming $\lim_{\tilde{n} \to \infty} W_{\Omega(t_{k\tilde{n}+s})} = L$, since (62) holds for every $z_{t^{*k}-s}$, substituting $z_{t^{*k}-s} = e_1^{\mathsf{T}} = (1, 0, \cdots, 0)^T$ in (62), we can prove that the first column of $L$ equals the first column of $W_{\Omega(t^{*k}-s)}$. Performing the same procedure for $z_{t^{*k}-s} = e_i^{\mathsf{T}}, i = 2, 3, \cdots, M$, yields

$$\lim_{\tilde{n} \to \infty} W_{\Omega(t_{k\tilde{n}+s})} = W_{\Omega(t^{*k}-s)}. \qquad (63)$$

According to (59), $U$ contains an infinite number of terms of the sequence $\{z_{t_n}\}_{n=1}^{\infty}$, i.e.

$$\exists N \in \mathbb{N} \ \ s.t. \quad n \geq N \implies z_{t_n} \in U. \qquad (64)$$

Suppose that $z_{t_n} \in U$ for some $n \geq N$. Thus, there exists some $s \in \{0, 1, 2, \cdots, k-1\}$ such that $z_{t_n} \in \{z_{t_{km+s}}\}_{m=1}^{\infty}$. Without loss of generality let $s = 0$. Hence, there is some $\tilde{n} \in \mathbb{N}$ such that

$z_{t_n} = z_{t_{k\tilde{n}}}$ and $\lim_{\tilde{n}\to\infty} z_{t_{k\tilde{n}}} = z_{t^{*k}}$. In this case, moving forward in time gives

$$z_{t_n} = z_{t_{k\tilde{n}}} \ \left(z_{t_n} \in \{z_{t_{km}}\}_{m=1}^\infty\right), \qquad\qquad \lim_{\tilde{n}\to\infty} z_{t_{k\tilde{n}}} = z_{t^{*k}},$$

$$z_{t_{n+1}} = z_{t_{k\tilde{n}+1}} \ \left(z_{t_{n+1}} \in \{z_{t_{km+1}}\}_{m=1}^\infty\right), \qquad\qquad \lim_{\tilde{n}\to\infty} z_{t_{k\tilde{n}+1}} = z_{t^{*k}-1},$$

$$z_{t_{n+2}} = z_{t_{k\tilde{n}+2}} \ \left(z_{t_{n+2}} \in \{z_{t_{km+2}}\}_{m=1}^\infty\right), \qquad\qquad \lim_{\tilde{n}\to\infty} z_{t_{k\tilde{n}+2}} = z_{t^{*k}-2},$$

$$\vdots$$

$$z_{t_{n+k-1}} = z_{t_{k\tilde{n}+k-1}} \ \left(z_{t_{n+(k-1)}} \in \{z_{t_{km+k-1}}\}_{m=1}^\infty\right), \qquad\qquad \lim_{\tilde{n}\to\infty} z_{t_{k\tilde{n}+k-1}} = z_{t^{*k}-(k-1)},$$

$$z_{t_{n+k}} = z_{t_{k(\tilde{n}+1)}} \ \left(z_{t_{n+k}} \in \{z_{t_{km}}\}_{m=1}^\infty\right), \qquad\qquad \lim_{\tilde{n}\to\infty} z_{t_{k(\tilde{n}+1)}} = z_{t^{*k}},$$

$$z_{t_{n+k+1}} = z_{t_{k(\tilde{n}+1)+1}} \ \left(z_{t_{n+k+1}} \in \{z_{t_{km+1}}\}_{m=1}^\infty\right), \qquad\qquad \lim_{\tilde{n}\to\infty} z_{t_{k(\tilde{n}+1)+1}} = z_{t^{*k}-1},$$

$$\vdots$$

$$z_{t_{n+2k-1}} = z_{t_{k(\tilde{n}+1)+k-1}} \ \left(z_{t_{n+2k-1}} \in \{z_{t_{km+k-1}}\}_{m=1}^\infty\right), \qquad\qquad \lim_{\tilde{n}\to\infty} z_{t_{k(\tilde{n}+1)+k-1}} = z_{t^{*k}-(k-1)},$$

$$z_{t_{n+2k}} = z_{t_{k(\tilde{n}+2)}} \ \left(z_{t_{n+2k}} \in \{z_{t_{km}}\}_{m=1}^\infty\right), \qquad\qquad \lim_{\tilde{n}\to\infty} z_{t_{k(\tilde{n}+2)}} = z_{t^{*k}},$$

$$\vdots \tag{65}$$

Consequently, for $n \geq N$ and $j \in \mathbb{N}$, we can write

$$\prod_{i=0}^{kj-1} \boldsymbol{W}_{\Omega(t_{n+kj-1-i})}$$

$$= \left(\prod_{i=1}^{k} \boldsymbol{W}_{\Omega(t_{k(\tilde{n}+j)+k-i})}\right)\left(\prod_{i=1}^{k} \boldsymbol{W}_{\Omega(t_{k(\tilde{n}+j-1)+k-i})}\right) \cdots \left(\prod_{i=1}^{k} \boldsymbol{W}_{\Omega(t_{k(\tilde{n})+k-i})}\right)$$

$$= \prod_{l=0}^{j} \prod_{i=1}^{k} \boldsymbol{W}_{\Omega(t_{k(\tilde{n}+j-l)+k-i})}. \tag{66}$$

On the other hand, in equation (28), there are different configurations for matrix $\boldsymbol{D}_{\Omega(t-1)}$ and hence different forms for matrix $\boldsymbol{W}_{\Omega(t_{k\tilde{n}+s})}$. In this case, the phase space of the system is divided into different sub-regions by some borders; see (Monfared & Durstewitz, 2020a;b) for more details. Also, since the system (28) is a linear map in each sub-region, the $k$ periodic points of $\Gamma_k$ must belong to different sub-regions (at least two different sub-regions). Accordingly, based on (63) and (65), there exists some $\tilde{N} \in \mathbb{N}$ such that for every $\tilde{n} \geq \tilde{N}$ both $z_{t_{k\tilde{n}+s}}$ and $z_{t^{*k}-s}$ belong to the same sub-region and so the matrices $\boldsymbol{W}_{\Omega(t_{k\tilde{n}+s})}$ and $\boldsymbol{W}_{\Omega(t^{*k}-s)}$ $(s \in \{0,1,2,\cdots,k-1\})$ are identical. Hence, for $n \geq N$, $\tilde{n} \geq \tilde{N}$ and $j \in \mathbb{N}$, equation (66) becomes

$$\prod_{i=0}^{kj-1} \boldsymbol{W}_{\Omega(t_{n+kj-1-i})} = \prod_{l=0}^{j} \prod_{i=1}^{k} \boldsymbol{W}_{\Omega(t_{k(\tilde{n}+j-l)+k-i})} = \left(\prod_{s=0}^{k-1} \boldsymbol{W}_{\Omega(t^{*k}-s)}\right)^j. \tag{67}$$

Therefore, similar to the part $(ii)$, we can prove for every $z_1 \in \mathcal{B}_{\Gamma_k}$, $\frac{\partial z_T}{\partial \theta}$ and $\frac{\partial \mathcal{L}_T}{\partial \theta}$ will also remain bounded. $\qquad\square$

### A.2.2 Proof of Theorem 2, Part (II)

*Proof.* $(ii)$ Let for every $T > 2$

$$\boldsymbol{L}_T := J_T^* J_{T-1}^* \cdots J_2^*. \tag{68}$$

$\{\boldsymbol{L}_T\}_{T\in\mathbb{N},\,T>2}$ is a sequence of matrices $\boldsymbol{L}_T = [l_{ij}^{(T)}]_{1\le i,j\le M}$ and, due to (13), $\lim_{T\to\infty}\|\boldsymbol{L}_T\| = \infty$. Hence, there is at least one sub-sequence $\{l_{mk}^{(T_n)}\}_{T_n\in\mathbb{N},\,T_n>2}$ (for some $m,k\in\{1,2,\cdots,M\}$) such that $\lim_{T_n\to\infty} l_{mk}^{(T_n)} = \infty$.

On the other hand

$$\frac{\partial \boldsymbol{z}_T^*}{\partial\theta} = \frac{\partial^+\boldsymbol{z}_T^*}{\partial\theta} + \sum_{t=1}^{T-2}\left(\prod_{r=0}^{t-1} \boldsymbol{J}_{T-r}^*\right)\frac{\partial^+\boldsymbol{z}_{T-t}^*}{\partial\theta}. \tag{69}$$

Moreover, there exists some $N>2$ such that (for $t = T - N + 1$)

$$\frac{\partial^+\boldsymbol{z}_{N-1}^*}{\partial\theta} \neq 0. \tag{70}$$

For $\theta$ as the $k$-th element of a parameter vector $\boldsymbol{\theta}$ (or belonging to the $k$-th row of a parameter matrix $\boldsymbol{\theta}$), the term

$$\left(\prod_{r=0}^{T-N} \boldsymbol{J}_{T-r}^*\right)\frac{\partial^+\boldsymbol{z}_{N-1}^*}{\partial\theta} \tag{71}$$

is a vector in which the $i$-th element is $l_{ik}^{(T)}\frac{\partial^+ z_{k,N-1}^*}{\partial\theta}$.

Since $\lim_{T_n\to\infty} l_{mk}^{(T_n)} = \infty$, due to (70) $\lim_{T_n\to\infty} l_{mk}^{(T_n)}\frac{\partial^+ z_{k,N-1}^*}{\partial\theta} = \infty$, which implies $\frac{\partial\boldsymbol{z}_T^*}{\partial\theta}$ will diverge as $T\to\infty$. Similarly, by (21), we can prove $\frac{\partial\mathcal{L}_T^*}{\partial\theta}$ is divergent for $T\to\infty$.

By Oseledec's multiplicative ergodic Theorem, the results also hold for every $\boldsymbol{z}_1\in\mathcal{B}_{\Gamma^*}$. □

### A.2.3 PROOF OF THEOREM 3

*Proof.* Let $\Gamma = \{\boldsymbol{z}_1,\boldsymbol{z}_2,\ldots\boldsymbol{z}_T,\cdots\}$ be a quasi-periodic attractor. Then, the largest Lyapunov exponent of $\Gamma$ is

$$\lambda = \lim_{T\to\infty}\frac{1}{T}\ln\|J_T\,J_{T-1}\cdots J_2\| = \lim_{T\to\infty}\frac{1}{T}\ln\left\|\frac{\partial\boldsymbol{z}_T}{\partial\boldsymbol{z}_1}\right\| = 0. \tag{72}$$

We prove for every $0<\epsilon<1$

$$\lim_{T\to\infty}(1-\epsilon)^{T-1} < \lim_{T\to\infty}\left\|\frac{\partial\boldsymbol{z}_T}{\partial\boldsymbol{z}_1}\right\| < \lim_{T\to\infty}(1+\epsilon)^{T-1}. \tag{73}$$

For this purpose, we show $\forall\, 0<\epsilon<1$

(I) $\lim_{T\to\infty}(1-\epsilon)^{T-1} < \lim_{T\to\infty}\left\|\frac{\partial\boldsymbol{z}_T}{\partial\boldsymbol{z}_1}\right\|$, and

(II) $\lim_{T\to\infty}\left\|\frac{\partial\boldsymbol{z}_T}{\partial\boldsymbol{z}_1}\right\| < \lim_{T\to\infty}(1+\epsilon)^{T-1}$.

Assume for the sake of contradiction that (I) does not hold. Then there exists some $0<\epsilon<1$ such that

$$\lim_{T\to\infty}(1-\epsilon)^{T-1} \ge \lim_{T\to\infty}\left\|\frac{\partial\boldsymbol{z}_T}{\partial\boldsymbol{z}_1}\right\|. \tag{74}$$

Therefore

$$\exists\, T_0>1 \ \ s.t. \ \ \forall\, T\ge T_0 \implies (1-\epsilon)^{T-1} \ge \left\|\frac{\partial\boldsymbol{z}_T}{\partial\boldsymbol{z}_1}\right\|, \tag{75}$$

and so

$$\exists T_0 > 1 \ \ s.t. \ \ \forall T \geq T_0 \implies \frac{\ln(1-\epsilon)^{T-1}}{T-1} \geq \frac{\ln\left\|\frac{\partial \boldsymbol{z}_T}{\partial \boldsymbol{z}_1}\right\|}{T-1}. \tag{76}$$

Consequently, due to (72), for $T \to \infty$ we have $\ln(1-\epsilon) \geq 0$. This implies $\epsilon \leq 0$, which is a contradiction.

Similarly if we assume (II) is not true, then there exists some $0 < \epsilon < 1$ such that

$$\lim_{T\to\infty} \left\|\frac{\partial \boldsymbol{z}_T}{\partial \boldsymbol{z}_1}\right\| \geq \lim_{T\to\infty} (1+\epsilon)^{T-1}. \tag{77}$$

Thereby

$$\exists T_0 > 1 \ \ s.t. \ \ \forall T \geq T_0 \implies \left\|\frac{\partial \boldsymbol{z}_T}{\partial \boldsymbol{z}_1}\right\| \geq (1+\epsilon)^{T-1}, \tag{78}$$

and thus

$$\exists T_0 > 1 \ \ s.t. \ \ \forall T \geq T_0 \implies \frac{\ln\left\|\frac{\partial \boldsymbol{z}_T}{\partial \boldsymbol{z}_1}\right\|}{T-1} \geq \frac{\ln(1+\epsilon)^{T-1}}{T-1}. \tag{79}$$

This means $\ln(1+\epsilon) \leq 0$ as $T \to \infty$, i.e. $\epsilon \leq 0$, which is a contradiction.

Therefore (14) holds for $\Gamma$ and also, according to Oseledec's multiplicative ergodic Theorem, for every $\boldsymbol{z}_1$ in the basin of attraction of $\Gamma$. □

### A.3 ADDITIONAL RESULTS ON RELATION BETWEEN DYNAMICS AND GRADIENTS

#### A.3.1 FURTHER RESULTS AND REMARKS RELATED TO THEOREM 2

**Remark 4.** *The result of Theorem 2 also holds for unstable orbits $\{\boldsymbol{z}_1, \boldsymbol{z}_2, \boldsymbol{z}_3, \cdots\}$ with positive largest Lyapunov exponent. Trivially, for such orbits that diverge to infinity (unbounded latent states) gradients of the loss function will explode as $T \to \infty$.*

**Remark 5.** *For RNNs with ReLU activation functions there are finite compartments in the phase space each with a different functional form. In such a case, to define the largest Lyapunov exponent of $\Gamma^*$, in the proof of Theorem 2 we assume that $\Gamma^*$ never maps to the points of the borders.*

Based on Theorem 2, we can also formulate the necessary conditions for chaos and diverging gradients in standard RNNs with particular activation functions by considering the norms of their recurrence matrix, for which the following Corollary provides the basis:

**Corollary 1.** *Let for a standard RNN*

$$\left\|diag\big(f'(\boldsymbol{W}\boldsymbol{z}_{t-1} + \boldsymbol{B}\boldsymbol{s}_t + \boldsymbol{h})\big)\right\| \leq \gamma < \infty. \tag{80}$$

*If the RNN is chaotic, then $\|\boldsymbol{W}\| \, \gamma > 1$.*

*Proof.* Assume for the sake of contradiction that $\|\boldsymbol{W}\| \, \gamma \leq 1$. From

$$\left\|\prod_{2<t\leq T} \boldsymbol{W} diag\big(f'(\boldsymbol{W}\boldsymbol{z}_{t-1} + \boldsymbol{B}\boldsymbol{s}_t + \boldsymbol{h})\big)\right\| \leq \prod_{2<t\leq T} \left\|\boldsymbol{W} diag\big(f'(\boldsymbol{W}\boldsymbol{z}_{t-1} + \boldsymbol{B}\boldsymbol{s}_t + \boldsymbol{h})\big)\right\|$$

$$\leq (\|\boldsymbol{W}\| \, \gamma)^{T-2}, \tag{81}$$

it is concluded that $\lim_{T\to\infty} \left\|\prod_{2<t\leq T} \boldsymbol{W} diag\big(f'(\boldsymbol{W}\boldsymbol{z}_{t-1} + \boldsymbol{B}\boldsymbol{s}_t + \boldsymbol{h})\big)\right\| < \infty$, which contradicts (13). This means $\|\boldsymbol{W}\| \, \gamma > 1$ is a necessary condition for the standard RNN to be chaotic. □

**Remark 6.** *For RNN with the tanh and sigmoid activation functions $\gamma = 1$ and $\gamma = \frac{1}{4}$, respectively. Thus, by Corollary 1, the necessary conditions for chaos in these two cases are $\|\boldsymbol{W}\| > 1$ and $\|\boldsymbol{W}\| > 4$, respectively.*

### A.3.2 OTHER CONNECTIONS BETWEEN DYNAMICS AND GRADIENTS

As sect. 3 elucidated, there is a direct link between the norms of the Jacobians of the RNN along trajectories and the EVGP. By observing this link, we can formulate some general conditions that will have implications for the behavior of the gradients regardless of the limiting behavior of the RNN, as collected in the following theorem:

**Theorem 4.** *Let $\mathcal{O}_{\boldsymbol{z}_1} = \{\boldsymbol{z}_1, \boldsymbol{z}_2, \ldots \boldsymbol{z}_T, \cdots\}$ be a sequence (orbit) generated by an RNN $F_{\boldsymbol{\theta}} \in \mathcal{R}$ parameterized by $\boldsymbol{\theta}$, and $\boldsymbol{P}_T := \boldsymbol{J}_T - \boldsymbol{I}, \ T = 2, 3, \cdots$.*

> *(i) Assume that $\mathcal{O}_{\boldsymbol{z}_1}$ is an orbit for which $\left\| \frac{\partial^+ \boldsymbol{z}_T}{\partial \theta} \right\| \leq \xi \ \forall t$. If $\sum_{T=2}^{\infty} \|\boldsymbol{J}_T\| < \infty$, then the Jacobian $\frac{\partial \boldsymbol{z}_T}{\partial \boldsymbol{z}_1}$, the tangent vector $\frac{\partial \boldsymbol{z}_T}{\partial \theta}$ and thus the gradient of the loss function, $\frac{\partial \mathcal{L}_T}{\partial \theta}$, will be bounded for $T \to \infty$.*

> *(ii) If $\sum_{T=2}^{\infty} \|\boldsymbol{P}_T\| < \infty$, then the Jacobian $\frac{\partial \boldsymbol{z}_T}{\partial \boldsymbol{z}_1}$ will neither vanish nor explode as $T \to \infty$.*

*Proof.* Let $\|.\|$ be any matrix norm satisfying $\|\boldsymbol{A}_1 \boldsymbol{A}_2\| \leq \|\boldsymbol{A}_1\| \|\boldsymbol{A}_2\|$.

$(i)$ By boundedness of $\frac{\partial^+ \boldsymbol{z}_T}{\partial \theta}$ we have

$$\left\| \frac{\partial \boldsymbol{z}_T}{\partial \theta} \right\| = \left\| \frac{\partial^+ \boldsymbol{z}_T}{\partial \theta} + \sum_{t=1}^{T-2} \left( \prod_{r=0}^{t-1} \boldsymbol{J}_{T-r} \right) \frac{\partial^+ \boldsymbol{z}_{T-t}}{\partial \theta} \right\|$$

$$\leq \xi \left( 1 + \sum_{t=1}^{T-2} \left\| \prod_{r=0}^{t-1} \boldsymbol{J}_{T-r} \right\| \right) \leq \xi \left( 1 + \sum_{t=1}^{T-2} \prod_{r=0}^{t-1} \|\boldsymbol{J}_{T-r}\| \right). \tag{82}$$

Moreover,

$$1 + \sum_{t=1}^{T-2} \prod_{r=0}^{t-1} \|\boldsymbol{J}_{T-r}\| \leq 1 + \sum_{p} \|\boldsymbol{J}_p\| + \sum_{p<q} \|\boldsymbol{J}_p\| \|\boldsymbol{J}_q\| + \sum_{p<q<r} \|\boldsymbol{J}_p\| \|\boldsymbol{J}_q\| \|\boldsymbol{J}_r\| + \cdots$$

$$= \left( 1 + \|\boldsymbol{J}_T\| \right)\left( 1 + \|\boldsymbol{J}_{T-1}\| \right) \cdots \left( 1 + \|\boldsymbol{J}_2\| \right) =: \prod_{t=2}^{T} \left( 1 + \|\boldsymbol{J}_t\| \right). \tag{83}$$

Since $\sum_{T=2}^{\infty} \|\boldsymbol{J}_T\|$ converges, according to (Wedderburn, 1964), the infinite products $\prod_{T=2}^{\infty} \left( 1 + \|\boldsymbol{J}_T\| \right)$ in (83) converge to a finite number $\tilde{\mathcal{K}} \neq 0$. Consequently, by (82) and (83)

$$\lim_{T \to \infty} \left\| \frac{\partial \boldsymbol{z}_T}{\partial \theta} \right\| \leq \tilde{\mathcal{K}} < \infty, \tag{84}$$

which implies $\frac{\partial \mathcal{L}_T}{\partial \theta}$ will be bounded for $T \to \infty$.

Furthermore

$$\lim_{T \to \infty} \left\| \frac{\partial \boldsymbol{z}_T}{\partial \boldsymbol{z}_1} \right\| \leq \prod_{T=2}^{\infty} \|\boldsymbol{J}_T\| := \lim_{T \to \infty} \left( \|\boldsymbol{J}_T\| \|\boldsymbol{J}_{T-1}\| \cdots \|\boldsymbol{J}_2\| \right) \leq \prod_{T=2}^{\infty} \left( 1 + \|\boldsymbol{J}_T\| \right) \leq \tilde{\mathcal{K}}, \tag{85}$$

which completes the proof.

$(ii)$ Since $\sum_{T=1}^{\infty} \|\boldsymbol{P}_T\| < \infty$, due to (Wedderburn, 1964) the infinite product

$$\prod_{T=2}^{\infty} \left( \boldsymbol{I} + \boldsymbol{P}_T \right) = \prod_{T=2}^{\infty} \boldsymbol{J}_T := \lim_{T \to \infty} \boldsymbol{J}_T \boldsymbol{J}_{T-1} \cdots \boldsymbol{J}_2, \tag{86}$$

converges to a matrix $\boldsymbol{K} \neq \boldsymbol{O}$, which implies

$$0 < \lim_{T \to \infty} \left\| \frac{\partial \boldsymbol{z}_T}{\partial \boldsymbol{z}_1} \right\| = \|\boldsymbol{K}\| < \infty. \tag{87}$$

$\square$

Part $(i)$ of Theorem 4 relaxes some of the conditions required in Theorem 1 for bounded gradients by imposing a Lipschitz condition on the immediate derivatives. Part $(ii)$ generalizes conditions satisfied, for instance, in orthogonal (unitary) RNNs (Arjovsky et al., 2016; Henaff et al., 2016) or fully regularized PLRNNs (Schmidt et al., 2021).

**Proposition 3.** *Let $\mathcal{O}_{z_1} = \{z_1, z_2, \ldots z_T, \cdots\}$ be an orbit generated by an RNN $F_\theta \in \mathcal{R}$ (parameterized by $\theta$), and $\|J_T\| \neq 0$, $T \geq 2$. If $\sum_{T=2}^{\infty} \ln \|J_T\|$ diverges to $-\infty$, then the Jacobian $\frac{\partial z_T}{\partial z_1}$ vanishes as $T$ tends to infinity.*

*Proof.* For $\|J_T\| \neq 0$, $T \geq 2$, we have

$$0 \leq \left\| \frac{\partial z_T}{\partial z_1} \right\| \leq \|J_T\| \, \|J_{T-1}\| \, \cdots \, \|J_2\| \; = \; e^{\ln \|J_T\|} \, e^{\ln \|J_{T-1}\|} \, \cdots \, e^{\ln \|J_2\|} \; = \; e^{\sum_{t=2}^{T} \ln \|J_t\|}. \tag{88}$$

Hence if $\sum_{T=2}^{\infty} \ln \|J_T\| \to -\infty$, then

$$\lim_{T \to \infty} \frac{\partial z_T}{\partial z_1} = O. \tag{89}$$

$\square$

### A.4 EMPIRICAL EVALUATION: DATASETS

**Lorenz attractor**  The Lorenz system (Lorenz, 1963) is a simplified model for atmospheric convection, given by

$$\begin{aligned} \frac{\mathrm{d}x}{\mathrm{d}t} &= \sigma(y - x), \\ \frac{\mathrm{d}y}{\mathrm{d}t} &= x(\rho - z) - y, \\ \frac{\mathrm{d}z}{\mathrm{d}t} &= xy - \beta z. \end{aligned} \tag{90}$$

The system is of particular interest for its chaotic regime and was studied here for $\sigma = 16$, $\rho = 45.92$ and $\beta = 4$. For these parameters the Lorenz system is known to have a maximal Lyapunov exponent $\lambda_{\max} = 1.5$ (Rosenstein et al., 1993). To generate a time series, the ODEs were integrated with a step size $\Delta t = 0.01$ using `scipy.integrate`. Accordingly, the prediction time is $\tau_{pred} = \frac{\ln(2)}{\Delta t \, \lambda_{max}} = 46.2$.

**Duffing oscillator**  The Duffing oscillator (Duffing, 1918) is an example of a periodically forced oscillator with nonlinear elasticity

$$\ddot{x} + \delta \dot{x} + \beta x + \alpha x^3 = \gamma \cos(\omega t). \tag{91}$$

Note that this system is non-autonomous, that is externally forced due to the r.h.s. of eqn. 91. The following parameters were chosen to arrive at a chaotically forced oscillator: $\alpha = 1.0$, $\beta = -1.0$, $\delta = 0.1$, $\gamma = 0.35$, and $\omega = 1.4$. For these parameters the Duffing oscillator has a maximum Lyapunov exponent of $\lambda_{max} = 0.0995$. The dataset used here was created with the code from (Gilpin, 2021) as a three dimensional embedding with step size $\Delta t = 0.17$. The prediction time is $\tau_{pred} = 39.28$.

**Rössler system**  Another prime textbook example for a chaotic system is the Rössler system (Rössler, 1976) given by:

$$\begin{aligned} \frac{dx}{dt} &= -y - z, \\ \frac{dy}{dt} &= x + ay, \\ \frac{dz}{dt} &= b + z(x - c). \end{aligned} \tag{92}$$

For the parameters $a = 0.15$, $b = 0.2$ and $c = 10$, the maximal Lyapunov exponent is $\lambda_{\max} = 0.09$ (Rosenstein et al., 1993). To arrive at a time series, a step size of $\Delta t = 0.1$ was chosen for integration. This gives us a prediction time of $\tau_{pred} = 77.0$ for this system.

**Mackey-Glass equation**  The Mackey-Glass equation (Glass & Mackey, 1979) is a nonlinear time delay differential equation

$$\dot{x} = \beta \frac{x_\rho}{1 + x_\rho^n} - \gamma x \quad \text{with } \beta, \gamma, \rho > 0. \tag{93}$$

Here $x_\rho$ represents the value of the variable $x$ at time $t - \rho$ (note that strictly, mathematically, this makes the system infinite-dimensional). Choosing the parameters to be $\beta = 2$, $\gamma = 1.0$, $n = 9.65$, and $\rho = 2.0$, leads to chaotic behavior with a maximum Lyapunov exponent of $\lambda_{max} = 0.21$. The dataset was created as a 10-dimensional embedding with the code from (Gilpin, 2021) using $\Delta t = 0.04$. This yields a prediction time of $\tau_{pred} = 82.2$.

**Empirical temperature time series**  This time series was recorded at the Weather Station at the Max Planck Institute for Biogeochemistry in Jena, Germany, spanning the time period between 2009 and 2016, and reassembled by François Chollet for the book *Deep Learning with Python*. The data set can be accessed at https://www.kaggle.com/pankrzysiu/weather-archive-jena.

To expose the underlying chaotic dynamics of the time series, trends and yearly cycles were removed, and nonlinear noise-reduction was performed (using `ghkss` from *TISEAN*, see also Kantz et al. (1993)). Fig. 4 (a) shows a snippet of the temperature data in comparison with the de-noised time-series. High-frequency noise was further reduced through Gaussian kernel smoothing ($\sigma = 200$), and the resulting time series was sub-sampled (every $5^{th}$ data point was retained). Fig. 4 (b) clearly reveals a fractional dimension of $D_{eff} = 2.8$ for the de-noised and smoothed time-series. This strongly suggests that the dynamics governing the time series are chaotic. We created a time delay embedding (Kantz & Schreiber, 2003) with $m = 5$ (estimated by the false nearest neighbor technique, see Kennel et al. (1992)) and delay $\Delta t = 500$ (obtained as the first minimum of the mutual information). The first three embedding dimensions are shown in Fig. 4(c). The maximal Lyapunov exponent of this time series was determined with `lyap_r` from *TISEAN* (Hegger et al., 1999) to be $\lambda_{\max} = 0.016$, see Fig. 3(a). This value is in close agreement with the literature (Millán et al., 2010). The predictability time of this system is estimated to be $\tau_{pred} = 43.3$.

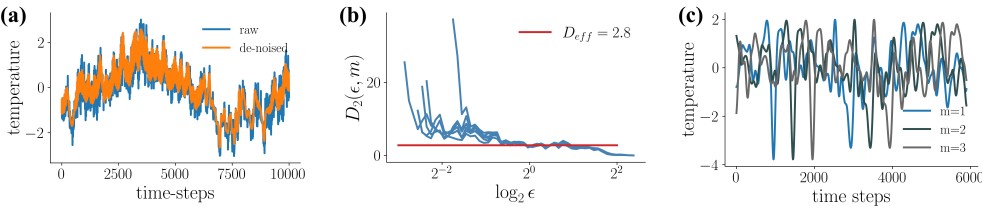

Figure 4: (a) Snippet of the original temperature data and de-noised time series. (b) Blue lines show the local slopes of the correlation sums for embedding dimensions $m \in \{5, \ldots, 10\}$. The convergence of these estimates in $m$ reveals a fractional dimension indicated by the plateau. (c) First three dimensions of the time-delay embedding series as used for training.

All datasets used were standardized (i.e., centered with unit variance) prior to training.

## A.5 EMPIRICAL EVALUATION: MEASURES OF RECONSTRUCTION QUALITY

**Attractor overlap**  To asses the geometrical similarity of the chaotic attractor produced by the RNN to the one underlying the observations, we calculate the Kullback-Leibler divergence of the ground truth distribution $p_{\text{true}}(\boldsymbol{x})$ and the distribution $p_{\text{gen}}(\boldsymbol{x}|\boldsymbol{z})$ generated by RNN simulation. To do so in practice, we employ a binning approximation (see (Koppe et al., 2019))

$$D_{\text{stsp}}\left(p_{\text{true}}(\mathbf{x}), p_{\text{gen}}(\mathbf{x} \mid \mathbf{z})\right) \approx \sum_{k=1}^{K} \hat{p}_{\text{true}}^{(k)}(\mathbf{x}) \log\left(\frac{\hat{p}_{\text{true}}^{(k)}(\mathbf{x})}{\hat{p}_{\text{gen}}^{(k)}(\mathbf{x} \mid \mathbf{z})}\right),$$

where $K$ is the total number of bins, and $\hat{p}_{\text{true}}^{(k)}(\mathbf{x})$ and $\hat{p}_{\text{gen}}^{(k)}(\mathbf{x} \mid \mathbf{z})$ are estimates obtained as relative frequencies through sampling trajectories from the observed time-series and the trained RNN, respectively.

**Power-spectral correlations** Since in DS reconstruction we aim to capture invariant (time-independent) properties of the underlying system, besides the geometrical agreement, we compare the similarity in true and RNN-reconstructed power spectra. To do so, we generate a time series of length $100,000$ from the RNN and calculate its power spectrum using the fast Fourier transform (scipy.fft). To reduce the influence of noise we apply Gaussian kernel smoothing and cut off the long high-frequency tails of the spectra. The dimension-wise correlation between observed and generated spectra are then averaged to give the $PSC$.

## A.6 FURTHER EMPIRICAL EVALUATIONS

### A.6.1 RECONSTRUCTION: RÖSSLER SYSTEM

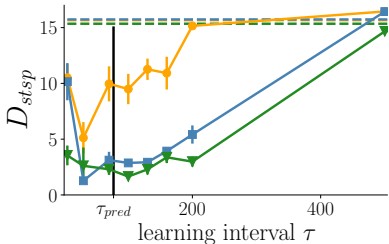 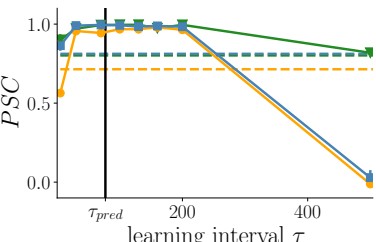

Figure 5: Overlap in attractor geometry ($D_{stsp}$, lower = better) and dimension-wise power-spectra correlations ($PSC$, higher = better) against learning interval $\tau$ for the Rössler attractor. Continuous lines = sparsely forced BPTT. Dashed lines = classical BPTT with gradient clipping. Prediction time indicated vertically in black.

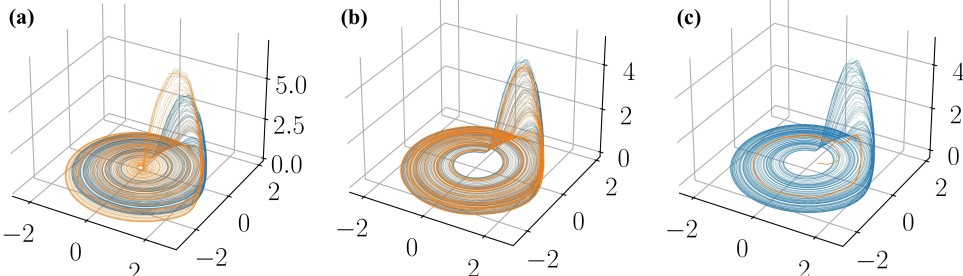

Figure 6: The Rössler attractor (blue) and reconstruction by a LSTM (orange) trained with a learning interval (a) chosen too small ($\tau = 5$), (b) chosen optimally ($\tau = 30$), and (c) chosen too large ($\tau = 200$).

### A.6.2 RECONSTRUCTION: HIGH-DIMENSIONAL MACKEY-GLASS SYSTEM

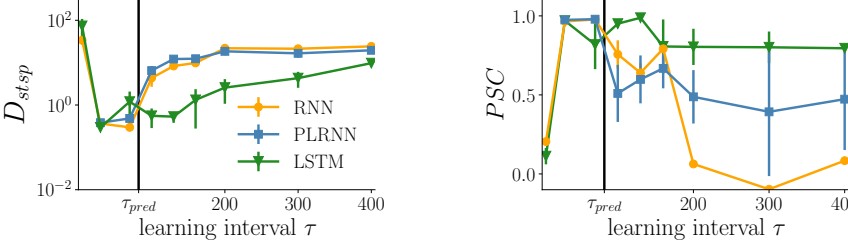

Figure 7: Overlap in attractor geometry ($D_{stsp}$, lower = better) and dimension-wise power-spectra correlations ($PSC$, higher = better) against learning interval $\tau$ for the 10d Mackey-Glass system. Continuous lines = sparsely forced BPTT. Dashed lines = classical BPTT with gradient clipping. Prediction time indicated vertically in black.

### A.6.3 RECONSTRUCTION: PARTIALLY OBSERVED LORENZ SYSTEM

For this evaluation we trained models only on the variables $\{y, z\}$ of the Lorenz system, eqn. 90. In order to compute the attractor overlap ($D_{stsp}$) in the true state space, however, after training the observation matrix $B$ was recomputed by linearly regressing the first 10 latent states onto the first 10 observations from all three Lorenz variables in eqn. 90.

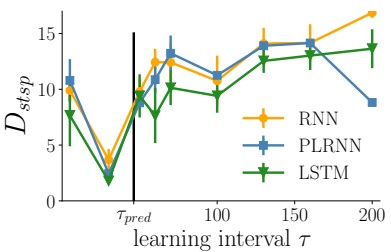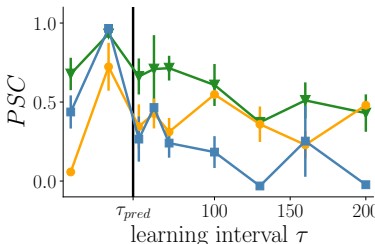

Figure 8: Overlap in attractor geometry ($D_{stsp}$, lower = better) and dimension-wise power-spectra correlations ($PSC$, higher = better) against learning interval $\tau$ for the partially observed Lorenz system. Continuous lines = sparsely forced BPTT. Dashed lines = classical BPTT with gradient clipping. Prediction time indicated vertically in black.

### A.6.4 OTHER INITIALIZATION PROCEDURES: STANDARD BATCHING WITH ZERO RESETTING

A common procedure in training RNNs is partitioning the time series into chunks (as we did based on the Lyapunov spectrum), but then simply resetting the hidden states at the beginning of each chunk (window) to 0. Formally this would mean that we do not force the trajectory back on track as in our approach, but instead may kick it off the track. To illustrate this, here we trained an LSTM on chunks (windows) with a length given by the optimal $\tau$ ($\tau_{opt} = 30$ for the Lorenz system), but then initialized the hidden states to 0 at the beginning of each window. The performance obtained this way is indicated by the green dashed line in Fig. 9 below.

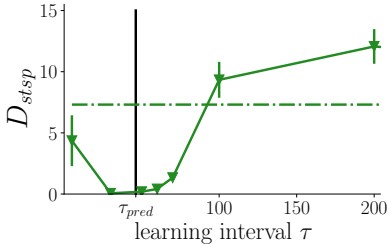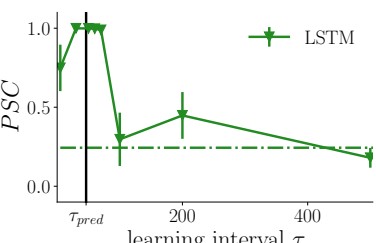

Figure 9: Overlap in attractor geometry ($D_{stsp}$, lower = better) and dimension-wise power-spectra correlations ($PSC$, higher = better) against learning interval $\tau$ for the Lorenz system. Continuous lines = sparsely forced BPTT. Dashed-dotted lines = windowing without forcing (choosing windows according to the optimal prediction time, but resetting hidden states to zero rather than its TF control value). Prediction time indicated vertically in black.

### A.6.5 ELECTROENCEPHALOGRAM (EEG) DATA

We used EEG data recorded by Schalk et al. (2004) and provided on PhysioNet (Goldberger et al., 2000), from which we took the baseline recording of the first patient for our analysis. Preprocessing was performed as outlined above for the temperature time-series, i.e. we applied nonlinear noise-reduction (see Fig.10 (a)) and Gaussian kernel smoothing ($\sigma = 30$). Fig. 10 (b) indicates a fractional dimension $D_{eff} = 1.8$ for the de-noised and smoothed times series. We created a time delay embedding with an embedding dimension of $m = 10$ and a delay time of $\Delta t = 500$. The maximal Lyapunov exponent for this time series was determined to be $\lambda_{max} = 0.007$, see Fig. 11 (a). To ease training, the time-series was sub-sampled and only every third data point was retained. With this, we obtain a predictability time $\tau_{pred} = 33.01$.

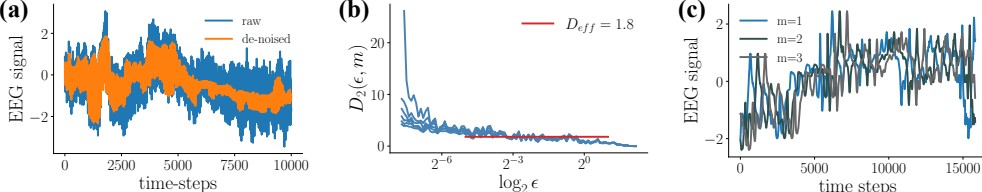

Figure 10: (a) Snippet of the original EEG data and de-noised time series. (b) Blue lines show the local slopes of the correlation sums for embedding dimensions $m \in \{5, \ldots, 10\}$. The convergence of these estimates in $m$ reveals a fractional dimension indicated by the plateau. (c) First three dimensions of the time-delay embedding series as used for training.

Full reconstruction of the dynamics from complex and noisy EEG signals is a very ambitious. Here we therefore focused on the system's short-term behavior and combined multiple shorter trajectory bits (of length $T = 70$) in the calculation of our measure for geometrical agreement in state space.

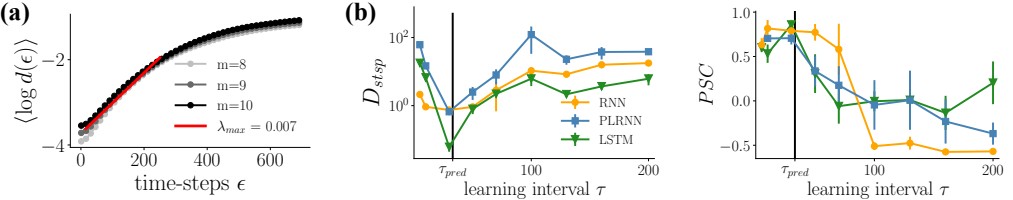

Figure 11: (a) The maximal Lyapunov exponent was determined as the slope of the average log-divergence of nearest neighbors in embedding space ($m = $ embedding dimension). (b) Reconstruction quality assessed by attractor overlap (lower = better) and power-spectrum correlation (higher = better). Black vertical lines = $\tau_{\text{pred}}$.

## A.7 SPARSELY FORCED BPTT

**Loss truncation** One implicit consequence of the teacher forcing, eqn. (16), is the interruption of the hidden-to-hidden connections at these time points. More specifically, if the system is forced at time $t \in \mathcal{T}$, then there is no connection between $z_t$ and $z_{t+1}$, that is

$$J_{t+1} = \frac{\partial z_{t+1}}{\partial z_t} = \frac{\partial RNN(\tilde{z}_t)}{\partial z_t} = 0. \tag{94}$$

To see how these vanishing Jacobians truncate the loss gradients w.r.t to some parameter $\theta$, let us focus on the loss gradients immediately after the forcing,

$$
\begin{aligned}
\frac{\partial \mathcal{L}_{t+1}}{\partial \theta} &= \frac{\partial \mathcal{L}_{t+1}}{\partial z_{t+1}} \sum_{k=1}^{t+1} \frac{\partial z_{t+1}}{\partial z_k} \frac{\partial^+ z_k}{\partial \theta} \\
&= \frac{\partial \mathcal{L}_{t+1}}{\partial z_{t+1}} \left( \frac{\partial^+ z_{t+1}}{\partial \theta} + \sum_{k=1}^{t} \underbrace{\frac{\partial z_{t+1}}{\partial z_k}}_{=0\,,\ \text{because of (94)}} \frac{\partial^+ z_k}{\partial \theta} \right) \\
&= \frac{\partial \mathcal{L}_{t+1}}{\partial z_{t+1}} \frac{\partial^+ z_{t+1}}{\partial \theta}.
\end{aligned}
\tag{95}
$$

Eqn. (95) shows that sparsely forced BPTT implicitly truncates the loss gradients because it interrupts the hidden-to-hidden connection from $z_t$ to $z_{t+1}$ for $t \in \mathcal{T}$. More generally, defining $\tilde{t} := \max\{t' \in \mathcal{T} : t' \leq t\}$, the overall loss gradients are truncated to

$$\frac{\partial \mathcal{L}}{\partial \theta} = \sum_{t=1}^{T} \frac{\partial \mathcal{L}_t}{\partial z_t} \sum_{k=1}^{t} \frac{\partial z_t}{\partial z_k} \frac{\partial^+ z_k}{\partial \theta}$$

$$\stackrel{\text{tr.}}{=} \sum_{t=1}^{T} \frac{\partial \mathcal{L}_t}{\partial z_t} \sum_{k=\tilde{t}}^{t} \frac{\partial z_t}{\partial z_k} \frac{\partial^+ z_k}{\partial \theta}. \tag{96}$$

