# OpenReview forum: "How to train RNNs on chaotic data?"
_ICLR.cc/2022/Conference — ICLR 2022 Submitted_

### Official Review · Reviewer_DWxP · 2021-11-01

**Correctness:** 2
**Technical Novelty And Significance:** 2
**Empirical Novelty And Significance:** 2
**Recommendation:** 5
**Confidence:** 4

**Details Of Ethics Concerns:**

No concerns.

**Main Review:**

Strengths: The authors establish a connection between Lyapunov exponents and behaviors of the dynamics of recurrent neural networks. Some derivations for particular recurrent neural networks are provided.

Weaknesses: In general, Lyapunov exponents are one of the most important and well-known techniques to characterize the dynamics of the chaotic systems. One of the concerns when training with time-varying signals the dynamics becomes essentially time-varying and the results as stated do not hold. Beside long calculations in the appendix, that are mostly straightforward I am not convinced that the approach will be numerically efficient. Lyapunov exponents are known to lead to primarily numerical computations that are very expensive. They also highly depend on initial conditions, and I failed to find any analysis related to this. In addition, the RNN dynamics are mostly bounded stable and show chaotic behaviors only in bounded regions and can be divergent only locally. Finally, I failed to find any comparison results with any other technique to illustrate and show the claimed effectiveness of the technique.


**Summary Of The Paper:**

The authors provide a study of gradients of the recurrent neural networks using Lyapunov exponents. They claim to offer a simple and effective training for the chaotic data. Some simulation results are provided.

**Summary Of The Review:**

Based on what is provided it is difficult for me to believe the claims are valid.

---

> ### Author Response · Authors · 2021-11-18
> **Response to point 1-4**
>
> *1) Time-varying signals*
>
> *All our results still hold with time-varying inputs*. We assume that by ‘time-varying’ the referee means a non-autonomous (externally forced) system (since a dynamical system by definition always describes an evolution in time, of course). Please note that none of our theorems in sect. 3 imposed any restrictions on external forcing (time-varying inputs), they all cover the non-autonomous case (eq. 1) as well. We further emphasize that, mathematically, non-autonomous (externally forced) systems can always be rewritten as autonomous dynamical systems (e.g. Perko 2001; Alligood et al. 1996). Whether one represents some variables through external inputs or treats them as dynamical variables in their own right, i.e. includes them as observations, is therefore mainly a matter of the specific scientific question asked, but none that makes a theoretical difference (see also added Remark 2). This is the reason why our procedure also works, for instance, on the real-world example, for which of course we have no knowledge about external inputs.
>
> To make this more explicit in the empirical section as well, we now have added a new example (new Fig. 1b) with time-varying external input.
>
> *2) Calculations in Appendix/ numerical efficiency in Lyapunov exponents*
>
> Some of the calculations are (luckily) relatively straightforward (which we would think is not per se a bad thing), but others rely on advanced concepts from topology and ergodic theory in the proofs of the theorems.
>
> The numerical costs for the Lyapunov exponents are fairly limited compared to the costs of doing a systematic hyper-parameter scan (and what else would be the alternative?). The actual computations only would need to be done once before training on the empirical data set, and usually take only a few seconds, after which $\tau$ could be fixed.
>
> More importantly, please note that the whole point of this paper is a systematic theoretical characterization of the relations between system dynamics (through Lyapunov spectrum) and gradients that has not been provided before. The empirical examples serve to illustrate the implications of the theory, and so they necessarily need to make a connection to the system’s Lyapunov spectrum. The results here are principled and general, whether one may be able or not to compute Lyapunov exponents in practice.
>
> *3) Dependence on initial conditions*
>
> This statement is not correct in this form. As cited in the text, there are theorems that guarantee that trajectories from *almost all* (in the sense of dense) initial conditions within an attractor’s basin of attraction will have the same maximal Lyapunov exponent (according to Oseledec’s multiplicative ergodic Theorem; see Ott, 2002). If we do not start within the basin of attraction of a considered attractor, then simply one of the other theorems applies and we need to consider a different limiting dynamics. In any case, there is no gap in our theoretical analysis with regards to this, the case of different initial conditions is completely covered. And note that the approach indeed worked out on all (now 6) dataset examples!
>
> *4) Chaos only in bounded regions*
>
> We didn’t quite understand this point or what it would imply. Of course, by the very definition of a chaotic attractor, the trajectories are bounded (otherwise you get the trivial case of unbounded dynamics, see Remark 4 in Appx. A.3.1). Now as soon as you train RNNs on a chaotic system like the Lorenz or Rössler attractor, or empirical data with these properties, then we precisely want it to capture exactly this dynamics (this is the whole point about DS reconstruction, otherwise we would not have captured the underlying system). But that immediately entails that the conditions of our Theorem 2 apply. We mathematically proved that under these conditions loss gradients will always diverge. Where exactly does the referee see the flaws or limitations in this?
>
> More generally, RNNs are formally discrete-time dynamical systems, and as such will exhibit any of the four limiting dynamics described in sect. 3 if released from any initial condition (see e.g. textbooks by Alligood et al. 1996, Perko 2001). We exhaustively described how the gradients will behave for each of these limiting cases, so one of them will always apply within any region of state space. In other words, we have comprehensively addressed all theoretically possible scenarios, there are no specific situations that we neglected as the referee’s comment seems to imply.

---

> > ### Author Response · Authors · 2021-11-18
> > **Response to point on comparison and lowly scored correctness**
> >
> > *5) Comparison with other techniques*
> >
> > This is a theoretical study. The whole point about the empirical section is to illustrate the theoretical implications (not to demonstrate the effectiveness of any specific technique). The theory says we need to consider the underlying system’s Lyapunov spectrum when training RNNs on time series observations, and this is exactly what we have demonstrated empirically *for a range of different RNN architectures* (plain RNNs, PLRNNs, LSTMs) and training procedures (including, besides various TF intervals, classical gradient clipping and batching approaches). We also showed that traditional remedies, like gradient clipping together with LSTMs, are not sufficient to address the problem. Which other techniques does the referee have in mind that we should have compared to here to make our points and illustrate the implications of the theory?
> >
> >
> > *“Difficult to believe claims”/ low correctness score*
> >
> > We provided *mathematical proofs* for our statements and theorems. When the referee states it’s difficult to believe the claims are valid, does s/he mean our proofs are wrong? Or does s/he mean the empirical results presented in Figs. 1-3 are technically flawed? In either case it would be helpful to know where exactly the referee thinks there are mistakes in our derivations or empirical evaluations, since otherwise we don’t have a chance to address her/his points.

---

### Official Review · Reviewer_BxLc · 2021-11-02

**Correctness:** 4
**Technical Novelty And Significance:** 3
**Empirical Novelty And Significance:** 3
**Recommendation:** 6
**Confidence:** 3

**Main Review:**

**Pros:**
- The paper provides a fresh look at how learning chaotic dynamics is difficult for established RNN architectures.
- The paper proposes to fix this issue by relying on existing architectures and techniques (teacher forcing), i.e., in contrast to proposing obscure architectures, which, unfortunately, seems to be standard in the research community nowadays.

**Weakness:**
- The Lorenz and Rossler systems used in the experiments seem to be deterministic and fully observable, i.e., the initial state + dynamics function defines the entire trajectory. Thus there is no need for using an RNN with a hidden state. One may directly learn a $x_{t} \mapsto x_{t+1}$ mapping using a feedforward network. What are the advantages of using an RNN, which suffers from gradient issues, compared to feedforward networks? Experiments on partially observable chaotic systems would have been appreciated.
- What happens in the "inverting the output mapping step" of the in teacher forcing when the output dimension is much lower-dimensional than the hidden state, and the output mapping is not uniquely invertible? Please elaborate.
- No code was provided

**Summary Of The Paper:**

The paper looks at the asymptotic behavior of the Jacobian of various RNN variants (standard, LSTM, GRU, PWL-RNN) when realizing stable vs chaotic dynamics. In particular, the paper shows that in the chaotic setting, the gradients asymptotically explode, i.e., when learning chaotic dynamics, the gradients have to explode (asymptotically).
The paper proposes to overcome this limitation by truncating the BP length and applying a teacher forcing method that periodically projects the observation onto the hidden state during training.

**Summary Of The Review:**

Overall solid paper. Please elaborate on the weaknesses mentioned above.

---

> ### Author Response · Authors · 2021-11-18
> **Response to 'weakness'**
>
> *1) Feedforward NNs/ partial observations/ noise*
>
> To use feedfoward NNs in this context (as generative models of the dynamics), after training one would need to recursively couple them to themselves by providing their last output as the new input (otherwise, unlike RNNs, they are not dynamical systems and hence cannot represent the dynamics of a DS). But this is mathematically equivalent to training a RNN with $\tau=1$ (since we use BPTT, i.e. we unroll the RNN across time as a FNN). This we already showed to be inferior to choosing an optimal forcing lag. Essentially, since a FNN only learns to couple directly consecutive time steps, it never sees the long-term behavior in the time series, and this is why $\tau=1$ indeed does not work well in our experimental evaluations.
>
> Furthermore, to learn a model of the dynamical system underlying empirically observed time series, in general it cannot be assumed that it could be approximated by a RNN of the same dimensionality. Almost always one needs to go to higher dimensionality, i.e. include hidden variables to accomplish a true embedding of the observed system (see theorems in Funahashi & Nakamura 1993, Kimura & Nakano 1998, Hanson & Raginsky 2020). Intuitively, part of the reason for this is that the RNN will usually have a very different functional form than the system that is being approximated.
>
> We followed the referee’s suggestion and included an example of a partially observed system in the revision (new Fig. 8).
> Re noise, we remark it is amply present, of course, in the real-world temperature time series (Fig. 3).
>
> *2) output dim < hidden dim/ invertibility*
>
> This is in fact the case in all our examples, but note we use the pseudo-inverse which worked well on all examples. One may also consider a further regularization factor in the pseudo-inverse as in ridge regression if one wants to avoid the invertibility problem altogether (added footnote 5).
>
> *3) Code availability*
>
> We have always provided all our code on our lab github site, and will definitely do so here as well.

---

### Official Review · Reviewer_rTPw · 2021-11-02

**Correctness:** 3
**Technical Novelty And Significance:** 2
**Empirical Novelty And Significance:** 2
**Recommendation:** 5
**Confidence:** 4

**Main Review:**

**Strengths**

- Analyzing chaotic behaviour of RNNs (vanilla, GRU, & LSTMs) based on the connection between loss gradients and Lyapunov spectrum is novel.

- Although ideas similar to sparsely forced BPTT exists in the literature, its interesting to see there exists a natural notion of the interval at which the teacher should be forced onto the student. Although its a little disappointing to note that its not easy to find such an interval and would live as a hyper-parameter in the algorithm.


**Weaknesses**

- Proposed sparsely forced BPTT provides feedback in the training process at regular intervals and such an idea has been explored in the literature ( see https://arxiv.org/pdf/1803.00144.pdf, https://proceedings.mlr.press/v139/kag21a.html, and references therein. )

- Paper claims that RNNs based on the fixed point idea or the restricted eigenspectrum of the recurrent matrix do not allow chaotic dynamics. No evidence for this point has been provided.


- In all the theorems, there have been assumption on what kind of dynamics is being followed by the RNN (be it fixed point, chaotic, quasi-periodic). One has no hold of any quantity that regulates the dynamics. In essence, the theory only characterizes how the gradients will behave knowing the dynamics and essentially under the hood just boils down to the eigenspectrum of the jacobians, which has been explored numerous times in the literature (starting with Bengio et. al. as to why gradients explode or vanish). This work does not help any practictioner to carefully choose one architecture over other given that the time-series dynamics is chaotic. Theoretical anaylsis given the dyanmics maybe one another way to characterise the EVGP and upto a certain extent has novelty in its own right.


**Questions for Authors**

- The predictability time for any other problem would indeed be treated as a hyper-parameter. Do you have any sense of what values might be a good place to initialize the search from?

- Sparsely forced BPTT looks a bit close in spirit to some alternative training schemes where auxiliary supervision is provided in the learning scheme (see https://arxiv.org/pdf/1803.00144.pdf, https://proceedings.mlr.press/v139/kag21a.html). Did the authors try out similar training scheme in their experiments?

- Is there any reason behind not using ODE-RNNs as a baseline, since they are designed to be operating with stable transitions?

**Writing Clarity**
- After Eq.1, For RNNs, you use $h_t$ to indicate the hidden states and the transition function $F_\theta$ for standard RNN, uses $h$ as the bias. Consider changing the symbol for bias for readability.
- Missing references
	- Practical Real Time Recurrent Learning with a Sparse Approximation ( https://openreview.net/forum?id=q3KSThy2GwB )
    - RNNs Incrementally Evolving on an Equilibrium Manifold: A Panacea for Vanishing and Exploding Gradients?  (https://openreview.net/forum?id=HylpqA4FwS)
	- Time Adaptive RNNs ( https://openaccess.thecvf.com/content/CVPR2021/papers/Kag_Time_Adaptive_Recurrent_Neural_Network_CVPR_2021_paper.pdf )
	- An Efficient Gradient-Based Algorithm for On-Line Training of Recurrent Network Trajectories ( https://ieeexplore.ieee.org/document/6797135 )
	- Learning Longer-term Dependencies in RNNs with Auxiliary Losses ( https://arxiv.org/pdf/1803.00144.pdf )
	- Training Recurrent Neural Networks via Forward Propagation Through Time (https://proceedings.mlr.press/v139/kag21a.html)

**Summary Of The Paper:**

This paper studies the exploding and vanishing gradient problem (EVGP) in the sequential modelling with chaotic dynamics. Theoretical analysis of the EVGP, based  on the relationship between loss gradients during RNN training and Lyapunov spectrum of RNN-generated orbits, is provided. Inspired by this analysis, an alternative to BPTT training algorithm is proposed, named sparsely forced BPTT, that forces the diverging dynamics to conform to the true trajectory, at regular time interval provided by the  Lyapunov spectrum.

**Summary Of The Review:**

Paper has two main contributions: (a) theoretical connections between loss gradients and Lyapunov spectrum that help in analyzing various RNN dynamics including chaos, and (b) sparse forced BPTT. The latter has been explored in the literature and the paper fails to mention these works in the literature review. Theoretical contribution is of some novelty, but it would be another way to explain EVGP, albeit with dynamics in mind.

---

> ### Author Response · Authors · 2021-11-18
> **Response to 'weaknesses'**
>
> *1) Novelty of training method*
>
> We do not consider the training algorithm to be our main contribution here (and have included the suggested refs. in Related Work), but still would like to stress that the modifications we introduced, although seemingly minor, were in fact *decisive* for the training success on DS reconstruction problems (some of our examples are otherwise just completely impossible with plain RNNs and standard BPTT!). The emphasis (and most of the workload), however, is clearly on the theoretical part. The main idea behind the empirical section was to illustrate practically the implications for the training process. Theoretically, a system’s Lyapunov spectrum is the central ingredient, and so we aimed to demonstrate this connection and its relevance for training in our empirical examples (to which now three new simulated and one new real-world example were added, see list above in general reply). This we think is indeed novel and important.
>
> *2) Evidence lacking that particular RNN designs are not sufficient*
>
> Regarding the eigenvalue spectrum (unitary and orthogonal RNNs), we indeed had provided a proof in Appx. A.1.5 (see also footnote 1).
>
> For coRNNs (Rusch & Mishra 2021a) and UnICORNNs (Rusch & Mishra 2021b) which both impose restrictions on the recurrence matrix, the original authors actually explicitly state this themselves (coRNN: see prop. E.1 and p.13; UnICORNNs: see pp.23, prop. C.1, in the original publications).
>
> For architectures that aim to enforce global fixed point dynamics (e.g. Antisymmetric RNNs and Lipschitz RNNs), by definition these will have a maximum Lyapunov exponent <0 (our Theorem 1), which rules out chaos by definition (you cannot have both, global convergence to a fixed point and chaos).
>
> If there is any other case we have forgotten, we are happy to address it.
>
> *3) Knowing the dynamics, dynamical scenarios considered*
>
> The referee seems to imply that our theory only considers very special cases which require specific assumptions to be met. But this is not at all the case, our theory is in fact completely generic and covers *all* types of dynamics that are theoretically possible in RNNs (which are formally discrete time dynamical systems; see textbooks by, e.g., Perko 2001; Guckenheimer & Holmes 1983; see also Arnold 1991, Springer). It does not make any restricting assumptions on the type of dynamics or external inputs that are allowed to occur. It is also not important to know the dynamics a priori: Mathematically, we know RNNs can only fall into one of the four considered classes, and we work out for each of them what will happen.
>
> Besides, empirically we actually do have established means to determine the dynamical regime through quantities like Lyapunov exponents or correlation dimensions (this is in fact what we did for our real-world example, see Figs. 3a, 4b; cf. also Kantz & Schreiber 2004). Please also note that we are specifically focussing on the chaotic case which is the one which really causes the trouble (i.e., in practise, there are not really that many different scenarios we need to consider).
>
> Moreover, the derivations do not boil down to mere Jacobians, as the referee suggests. Rather, it is a system’s Lyapunov spectrum that matters. While Jacobians naturally occur in the definition of the Lyapunov exponents, there are important and mathematically crucial differences, since we need to consider the behavior along full orbits of the system, whether periodic (cycle) or even aperiodic (the chaotic case).
>
> This is therefore not just another way to characterize the EVGP; rather we – for the first time as far as we are aware of – point out a *principle problem in training RNNs on chaotic time series* that cannot be avoided. This has important implications for RNN design and training: We cannot generally fix diverging gradients, as it has often been attempted in the past, just by imposing certain architectural or matrix constraints (see reply above), at least as long as we would like to address DS reconstruction problems. Rather, we should pay more attention to a careful design of the training process itself. We thought this is a very important take-home that needs to be acknowledged in further research!
>
> We have added and discussed the suggested references.

---

> > ### Author Response · Authors · 2021-11-18
> > **Response to questions**
> >
> > *Predictability time*
> >
> > One can always try to numerically estimate Lyapunov exponents as we have done for our real-world example, and one only needs to do this *once* before starting training and then fix $\tau$. In fact, in our case (Fig. 3a) this worked perfectly well, although temperature data are known to be quite tricky and noisy. Even if there is not a clear log-linear scaling region in the divergence plot (see Fig. 3a), a linear fit should still provide a reasonable first estimate.
> >
> > *Other training schemes*
> >
> > Thanks for pointing out (now cited & discussed). But please bear in mind that our main goal here was not to provide an all-new training scheme (we acknowledged that TF has been around before), but rather mainly use it as a vehicle to demonstrate the implications of the theory. As pointed above, however, the modifications to the training process we introduced were still crucial and made a huge practical difference on the problems we studied.
> >
> > *ODE-RNNs*
> >
> > ODE-based systems just like discrete-time RNNs can (and will, if not specifically constrained) exhibit all the different types of dynamics exposed in the paper (see textbooks by Perko 2001, Alligood et al. 1996). ODE-based RNNs in general have nothing built in a priori that would prevent exploding gradients when trained on chaotic data (they will have positive maximal Lyapunov exponents, just as their discrete time cousins, and, moreover, by any numerical solver would effectively be solved as discrete time maps). Special types of ODE-formulations (e.g. coRNN, UnICORNN) are designed to prevent divergence, but - as we pointed out above - this comes at the cost that chaos is not possible (qua design, as completely acknowledged by the authors themselves).
> >
> > We also would like to emphasize again that we see this as a theoretical study, we are not aiming for a new SOTA, but wanted to work out principle problems that so far have been largely neglected but need to be considered in the design of any algorithm (including ODE-RNNs).

---

> > > ### Author Response · Authors · 2021-11-18
> > > **Response to summary evaluation**
> > >
> > > (a) The theoretical contribution is not just another way to explain the EVGP. Rather, it adds a completely new perspective to it that – as far as we are aware – has not been considered anywhere before: Previous work always worked from the assumption that diverging gradients need to be avoided by enforcing certain Jacobians, while here we conclude it *cannot* be avoided as a matter of principle on many important problems, and we actually need to find strategies to deal with this. Furthermore, we offer a comprehensive mathematical theory linking gradients and dynamics under all theoretically possible scenarios, extending beyond those that have been considered before (fixed points and simple cycles). We are not aware the links to chaos (or quasi-periodicty) have been discussed before?
> > >
> > > (b) We have added this literature, but - as already stressed above - our contribution here is neither BPTT nor TF of course (and we didn’t claim so in our text). We mainly used these training schemes to exemplify the theoretical implications, and this explicitly demonstrated relation to the Lyapunov spectrum to our knowledge is indeed something new. Nevertheless, the dynamical systems perspective also led to amendments of the existing training algorithms that may seem minor at the surface, but turned out to be profound for the training success.

---

### Official Review · Reviewer_F1yn · 2021-11-04

**Correctness:** 4
**Technical Novelty And Significance:** 2
**Empirical Novelty And Significance:** 2
**Recommendation:** 6
**Confidence:** 4

**Main Review:**

## Pros:
* The paper is very clear and well written
* I very much like the idea of providing a systematic and general overview to analyse the gradient behavior of RNNs
* Very thorough literature review


## Concerns:

* In reality, an RNN trajectory heavily depends on the external input (forcing). The paper also states in the very end of the discussion that this forcing can significantly change the dynamics. Hence, it would very interesting to see some sensitivity analysis with respect to changes in the input for popular architectures (one with stable fixed points and one with limit-cycle behavior for instance) and a discussion on how this might change the provided theory.

* My biggest concern is regarding the experiments. Given the cut-off interval $\tau$, I understand the suggested approach to be equivalent to simply splitting the input sequence into smaller windows of length $\tau$ (in a mini-batch manner), and setting the initial hidden-state(s) to the pseudo-inverse of the ground-truth of the last entry of the window before. Hence, the only contribution is using specific initial hidden states and a theoretically motivated length for the windows (which in most cases has to be found anyway by hyperparamter optimization, as in many real-world applications the max. Lyapunov exponent cannot be efficiently computed). I would like the paper to acknowledge that the provided approach is in fact equivalent to windowing, which is a very standard practice for training RNNs. Moreover, I would like to see results using the same window length $\tau$, as in the suggested approach, but using a standard initial hidden state (set to zero). I suspect, that the results will be quite similar to the reported results of the suggested method (given that LSTMs for instance have been very successfully applied on chaotic time series prediction). But I could be mistaken.

* It feels very natural to choose the predictability time to be the length of the intervals $\tau$. However, following the reference provided in the paper (Bezruchko \& Smirnov, 2010), predictability time is a way more complex concept and can even be defined without appealing to the Lyapunov exponent, meaning systems with the same max. Lyapunov exponent can have very different predictability times. Therefore my question: Why was the predictability chosen as $\tau_{\text{pred}}=\ln(2)/\lambda_{max}$? Was it simply a fit to the provided experiments? In my opinion, more experiments need to be done in order to have a valid empirical support for that, e.g. what about a high-dimensional chaotic systems in say $>50$ dimensions. I suspect that the chosen predictability time won't be an accurate choice anymore.

* It would be interesting to compare the suggested training method to other approaches suggested in the context of training RNNs (for instance LSTMs) on chaotic systems, e.g. (Vlachas et al., 2018) in the paper.

* Unfortunately, the empirical investigation is very limited and I would have liked to see results on other tasks, such as NLP tasks, health-care applications and others (as paper claims in the introduction: chaotic behavior can be found in many sequential data sets), to see if this approach actually increases the performance of RNNs in widely-used RNN applications.
    That being said, language modeling (such as the Penntree bank (PTB) word-level and char-level task) would have been an excellent application, as the training of the models is already based on cutting the very long tokens stream into smaller windows, but in contrast to the suggested method by the authors, the initial hidden states are not pseudo-inverse of the ground-truth but are set to the last hidden state of the window before.

### Minor comments:
* The pseudo-inverse introduced in eq. (15) needs more discussion: Why can it be constructed like that (i.e. assumption of linearly independent rows)?
* Please define things like basin of attraction (first used in Theorem 1)


**Summary Of The Paper:**

The paper connects the gradient behavior of recurrent neural networks with its dynamics through the maximum Lyapunov exponent of a trajectory, which is generated by the respective RNN. In particular, several connections between RNNs which produce specific dynamics (e.g. with stable fixed points or limit-cycles) and the mitigation of the exploding gradient problem are proved. Moreover, the paper shows that RNNs generating a chaotic trajectory always suffer from the exploding gradient problem.
That being said, the paper emphasizes that in order to approximate chaotic systems, the RNN has to be able to produce chaotic trajectories, which is ruled out by design by several state-of-the-art RNNs that are constructed to solve the exploding and vanishing gradient problem. However, given that these RNNs - at least the moment the optimizer finds parameters such that the resulting RNN produces chaotic trajectories - will always run into the exploding gradient problem, the paper suggests an algorithmic fix for that, by forcing the hidden-states of the RNN to be the pseudo-inverse of the underlying ground-truth trajectory at equidistantly distributed points in time during training. This cuts off the gradients at these points in time and forces the output states to be close to the ground-truth trajectories. The interval for that is chosen based on the maximum Lyapunov exponent.
In the paper, three experiments are provided, where two of them are based on approximating a chaotic dynamical system (Lorenz and Rössler), while one is based on real-world data from temperature forecasting.

**Summary Of The Review:**

At this point, I cannot vote for acceptance, as

* the provided theory is helpful to get an overview over the various general ideas of tackling the exploding and vanishing gradient problem, however, it is often prohibitively hard to prove (or it is simply not true) that the RNN exhibits specific dynamics (for instance limit-cycle behavior), due to the role of the external input, for which preferably no prior assumption should be made. That being said, the provided framework might be very limited in its usefulness, when it comes to constructing new RNN architectures which mitigate the exploding and vanishing gradient problem.
* the suggested training method seems to be a minor deviation of very standard approaches
* the empirical evidence is not sufficient (more diverse datasets should be considered) and the presented results lack more useful comparisons.

I am happy to increase my rating, if the authors can resolve my concerns.


=====POST-REBUTTAL COMMENTS========

In general, I'm satisfied with how the authors addressed my minor concerns, in particular I appreciate the additional experiments.\
However, my main concern that the theoretical framework provided is very limited in its usefulness remains.\
I raise my score to (very) marginal acceptance.

---

> ### Author Response · Authors · 2021-11-18
> **Reply to point 1-4**
>
> *1) No external input*
>
> As pointed out in our general reply, there is a basic misunderstanding here: None of the theorems in sect. 3 requires the dynamics to be autonomous, our theory applies whether there is external input or not (i.e., regardless of what causes the chaos). We also emphasize that, mathematically, non-autonomous (externally forced) systems can always be rewritten as autonomous dynamical systems (e.g. Zhang et al. 2009, Springer;  Perko 2001; Alligood et al. 1996). In fact, whether one represents some variables through external inputs or adds them directly to the dynamical system as dynamical variables in their own right is mainly a matter of the specific scientific question asked.
>
> For the empirical evaluation (to which the statement in the Discussion referred to), it is true we focused on the case without explicit external inputs. This is not at all a rare situation, but common in scientific applications where commonly just time series are observed (e.g. in climate physics or ecology), but no external perturbations are (or can be) applied. But perhaps there was also another misunderstanding here: In the considered scientific contexts, RNNs are trained in an unsupervised way, i.e. what is often treated as input (e.g. when training RNNs in NLP) is treated as observations here. This is also why this approach indeed works on the temperature data, although we in fact have no knowledge about external inputs here!
>
> Nevertheless, to fully settle this point, we now have also added an example with explicit external forcing in addition to what we call the observations (new Fig. 1b).
>
> *2) Similarity to windowing, standard initial condition*
>
> We are happy to point out the relation to windowing (now added to Related Work and sect. 4). While TF is not equivalent to just windowing (see below), please also note that we do not claim this is our major contribution here. Our main contribution is clearly the theoretical part, and we used the procedure in sect. 4 mainly to demonstrate empirically the theoretically proven relation between Lyapunov spectrum and learning dynamics. This is the empirical validation that the “optimal window” (TF lag in our terminology) will crucially depend on the dynamics of the system, in a way that can be theoretically explained, motivated, and constrained, and has not been pointed out before.
>
> Re-initializing the latent states with zeros instead through the empirically observed values will accomplish the opposite of what we would like to achieve, namely kicking the trajectory *off track* instead of pulling it back. We have now added experiments (new Fig. 9) which explicitly demonstrate that this doesn’t work, i.e., the results are indeed much worse, as one would theoretically expect.
>
> Thus, taken together, *both* the forcing and the optimal training lag are crucial (and it actually took us a while to find out, training to reconstruct a DS is in fact much harder than just prediction!). This shows that a DS perspective can lead to new insights how, when and why RNN training approaches may work or fail.
>
> *3) Predictability time in high-dim. systems*
>
> The major purpose of the empirical section was to demonstrate the implications of the theory, which means a relation with the system’s Lyapunov spectrum, and that we think we achieved. Please note that the discussion in (Bezruchko & Smirnov, 2010), to which the referee refers, refers to the case where different initial conditions are associated with different Lyapunov exponents. However, here we explicitly consider the case where we would like to learn an underlying chaotic DS (otherwise the problem would not arise in the first place!): Within the basin of a (chaotic) attractor, due to Oseledec’s multiplicative ergodic Theorem (Ott, 2002), and more generally if there is a natural measure on the attractor (Ott, 2002), nearly all initial conditions will have the same max. Lyapunov exponent. In other words, the situation discussed in (Bezruchko & Smirnov, 2010) is not relevant here, so that the formulation we use is indeed valid and optimal. This is indeed confirmed by all empirical examples we have discussed.
>
> Either way, we followed the referee’s suggestion and also probed this approach on a high-dimensional chaotic benchmark now (high-dimensional Mackey-Glass equations; see new Fig. 7).
>
> *4) LSTMs for comparison*
>
> Please note that we indeed had tested this for LSTMs (see Figs. 1 \& 3, green curves)! As Figs. 1 \& 3 illustrate, for LSTMs precisely the same considerations based on the Lyapunov spectrum apply as for any of the other RNNs, and as predicted by the theory (see also Appx. A.1.3).

---

> > ### Author Response · Authors · 2021-11-18
> > **Limited empirical investigation**
> >
> > We provided an extensive theoretical coverage with many new theorems, proofs (some of which relying on advanced concepts from topology and ergodic theory), and important insights for training that haven’t been discussed in the literature before. Developing such theory and assuring its mathematical correctness is a time-intense undertaking by itself, and we have the impression this was considerably undervalued in some of the reviews.
> >
> > The empirical section was intended to illustrate practically some implications of this theory for RNN training. It does so on three different examples (now six!) which is fairly above the standards for theoretical papers, and more than common even in many algorithmic/ applicational papers which usually discuss just 2-4 datasets/ benchmarks (please compare to other ICLR publications!). The empirical examples come from a broad area in its own right, scientific time series and dynamical systems reconstruction, where RNNs are widely applied (whole ICLR, ICML, or NeurIPS papers are just dealing with this, see our Related Work). Even many more algorithmically oriented papers rarely provide a comprehensive survey of data types, but focus on a reasonable subset close to the motivation of the original development. Consider as an example the paper by Vaswani et al. (2017) on Transformers: It just considered two benchmarks (machine translation and parsing), both only in the language domain!
> >
> > In particular, language data would open a completely new chapter: They are very different from time series we most commonly encounter in science, and calculating Lyapunov exponents for them alone is a whole story (paper) in its own right. This would go far beyond what would be reasonable to expect within the scope of a single paper, it would add a whole new package which may make for a standalone empirical investigation and would take the theoretical focus of our work away.
> >
> > We hope the referee finds these considerations reasonable.
> >
> > This being said, please note that we actually have added three more examples (see list above, new Figs. 1b, 7, 8). With this, we definitely already provide more in terms of an empirical evaluation than most (published) papers in the field.

---

> > > ### Author Response · Authors · 2021-11-18
> > > **Minor comments**
> > >
> > > *Pseudo-inverse*. The choices here were really just for convenience. One could add, e.g., a regularization term as in ridge regression to alleviate invertibility issues (see added footnote 5). But empirically we didn’t run into problems anyway on all examples considered, so we didn’t feel pressed to explore much beyond this.
> > >
> > > *Basin of attraction* now defined in footnote 3.

---

> > > > ### Author Response · Authors · 2021-11-18
> > > > **Reply to summary evaluation**
> > > >
> > > > *Limitations of theory*. As pointed out above and in our general reply, the theory doesn’t make any such prior assumptions. The case of external inputs is completely covered by our theory (and now also empirically, new Fig. 1b). Moreover, the dynamical scenarios discussed are exhaustive, the limiting behavior of RNNs will always fall into one of the discussed classes (see, e.g., Perko 2001; Guckenheimer & Holmes 1983). Our theory is therefore not restricted to special cases, but completely generic.
> > > > Please also note that we can indeed assess the dynamical regime (limit cycle, chaos etc.) also empirically (cf. Kantz & Schreiber 2004): This is what in fact we have done for the temperature data through numerical determination of Lyapunov exponents (Fig. 3a) and the correlation dimension (Figs. 4b).
> > > >
> > > > Most importantly, our main point here focuses on the chaotic regime: As soon as the time series for training come from a chaotic system (which will be the rule rather than the exception, and which is really all we would need to know), then - as we prove - the exploding gradient problem cannot be avoided as a matter of principle. This is a very important point: It implies one needs to think along completely different lines than many current attempts to address training on long-term problems, for which we suggested directions empirically. We believe these are indeed highly relevant and useful insights.
> > > >
> > > > *Training method minor deviation*. First, TF and the optimal training lag turned out to be *decisive* modifications (without them it was hardly possible to train any architecture for DS reconstruction by just BPTT). But more importantly, our paper is not about a “groundbreaking” new training method. Rather, it is an extensive and detailed theoretical study of a long-standing and important problem. The training algorithm is secondary, it just serves to work out the theoretical points. These, however, we think are indeed very novel and impactful: Note that in all our examples the performance really profoundly depends on the TF lag in a way that is theoretically motivated.
> > > >
> > > > *Empirical sect. too limited*. With the newly added examples our paper now discusses more datasets than even most algorithmic/architecture papers would typically do, most certainly more than we have seen in any other theoretical study (please compare to other ICLR papers). It may be that this referee comes from a different area where language examples are more relevant, but for the area of science the examples we discussed are hugely relevant and are indeed very common benchmarks in this literature.

---

> > > > > ### Comment · Reviewer_F1yn · 2021-11-26
> > > > > **Reply to rebuttal**
> > > > >
> > > > > I appreciate that the authors added a few more empirical details. However, I still have some concerns, which were not adequately addressed:
> > > > >
> > > > > * Discussion around input in the theory: The authors might have misunderstood me (and possibly other reviewers who were pointing out similar issues). I don't criticize that external input is missing in the theorems (as correctly pointed out by the authors it is indeed covered). The main issue is the following: Given your theoretical findings, one would like to construct a new RNN and would like to apply your theory in order to say something about the exploding/vanishing gradient problem for this particular new RNN.
> > > > > For the exploding gradient problem: Wouldn't one still need to prove global stability in order to apply your theory? (that being said without any assumptions on the input, as well as parameters on the RNN!). This however is far from trivial and probably in most cases not possible (or simply not true).
> > > > > For vanishing gradient problem: One would need to show that the dynamics has no stable fixed point (for any choice of parameters and inputs). Which again is probably only possible under very strict constraints on the RNN design.
> > > > > I hope this detailed explanation makes my concerns regarding input/forcing clearer.
> > > > >
> > > > > * Showing that chaotic behavior leads directly to exploding gradients is arguably an important point (main contribution in this paper). However, since the full gradient of the RNN does also depend directly on the hidden states, doesn't a chaotic behavior directly imply exploding full gradients and looking at the partial derivatives is not necessary, as the full gradient anyway already explodes?
> > > > >
> > > > > * I appreciate that the authors added the windowing with zero hidden initialization to the experiments. It is, however, very surprising to me that there is apparently no deviation in performance when applying this approach to LSTMs. Given my own experience and the nature of chaotic systems, I would expect to see quite a bit deviation in performance for different window lengths. That being said, it is very hard to judge the quality of the empirical work without being able to look at the code. Can the authors explain what kind of hyperparameter optimization they have done in this case, especially in contrast to the other results.

---

> > > > > > ### Author Response · Authors · 2021-11-28
> > > > > > **Purpose of dynamical systems reconstruction & external inputs**
> > > > > >
> > > > > > It’s evident that there are some profound and deep misunderstandings here, maybe even reaching beyond our specific paper, about the field of dynamical systems and DS reconstruction more generally.
> > > > > >
> > > > > > We will try our best to clear this up and address the referee’s points:
> > > > > >
> > > > > > (1) First, just for clarification, we would like to point out that mathematically the terms ‘forcing’, or equivalently ‘external inputs’, are very well defined: Forcing refers to time-dependent functions in the formulation of a DS that are not part of the dynamics themselves, like the $s_t$ terms in the generic map eq. (1) or the $cos(\omega t)$ term in the definition of the forced Duffing oscillator, eq. (91). We really just used the common textbook definitions here (see list of references above), so we thought that’s what the referees had in mind as well when they spoke about forcing and inputs.
> > > > > >
> > > > > > But the real misunderstanding from what we understood now is the following: The referee seems to believe that we would *need to construct a RNN to ensure (or avoid) a certain type of dynamics*, like fixed point or chaotic dynamics. This could in principle be done, but it is neither the goal nor would it address the real problem.
> > > > > >
> > > > > > Rather, convergence to fixed points, cycles, or chaotic attractors are *properties of the real physical or biological system that underlies the actually observed data* (they are given by the natural system under investigation). For instance, for the temperature data in Fig. 3 we established through estimation of the system’s Lyapunov exponent and fractal dimension that these come from a chaotic DS (this is a common approach, see e.g. textbook by Kantz & Schreiber 2004).
> > > > > >
> > > > > > Now, if we want to *reconstruct* this DS underlying the data, the system we are using for approximation (RNNs in our case) *must have* the very same dynamical properties *after* training – it must express chaos, otherwise it would *not be a model of the underlying chaotic DS*! (Note, for instance, that the example reconstructions in Fig. 2b or Fig. 6b are not mere predictions of an RNN, but the RNN – *after training* – exhibits the very same dynamical behavior as the true DS when *simulated* from any initial condition.)
> > > > > >
> > > > > > This is where Theorem 2 comes in: It tells us that exploding gradients are *mathematically inevitable* when we wish to reconstruct a data-generating DS with chaotic dynamics. Once we understood this fundamental mathematical point, the implication is that we need to think of different ways of successfully training an RNN on chaotic data: We cannot simply fix the problem by architectural design (we cannot avoid it, as *a matter of principle*). Rather, we need to modify the training process itself in such a way that long time scales are still represented yet exploding gradients are avoided. This is what we attempted to illustrate in sect. 4.
> > > > > >
> > > > > > Does this resolve the misunderstanding?
> > > > > >
> > > > > >
> > > > > > (2) Our theorems and proofs consist of two parts: The (i) first refers to the explosion (or vanishing) of the Jacobians $\partial z_T/\partial z_t$ connecting two temporally distant states, and the (ii) second considers the explosion (vanishing) of the loss gradients $\partial L/\partial \theta$. Mathematically, divergence of loss gradients (ii) *does not* imply divergence of Jacobians (i)! This is why we distinguished these cases; for instance, the series in eq. (56) or eq. (69) may diverge while the Jacobians do not.
> > > > > > Does this answer the question?
> > > > > >
> > > > > >
> > > > > > (3) In our understanding, the referee had requested to rerun the training using the very same window length that led to optimal reconstructions in the TF approach, but to reinitialize the states to zero rather than to an initial value inferred from the data for each window. And that’s exactly what we did (see Fig. 9 legend: We have indicated the resulting performance value for that optimal lag as a long dashed line, rather than a single point at the optimal lag, perhaps that’s what was confusing?). Theoretically this is also exactly what one would expect. Perhaps the referee is referring to a different class of tasks or problems, e.g., one where either the underlying dynamics is not chaotic, or one which does not involve DS reconstruction but perhaps a sequence regression or simple prediction problem?

---

> > > > > > > ### Comment · Reviewer_F1yn · 2021-11-28
> > > > > > > **reply**
> > > > > > >
> > > > > > > I think there is still a misunderstanding between the authors and the referee: The intention of this question is my concern of extending your theoretical framework to more general use-cases of RNNs, e.g. constructing a new RNN and train on seqMNIST for instance and understanding the gradient dynamics of this RNN then thanks to your theoretical framework.
> > > > > > >
> > > > > > > That being said, I would have very much preferred to talk about this point and as a side-note: I find it a bit inappropriate to speculate about the expertise/scientific background of the referees instead of only focusing on the raised concerns. The point that this framework cannot easily be used/extended to the standard use-cases of RNNs (as described now several times) is one major limitation I and, as far as I understood, also other referees tried to point out.
> > > > > > >
> > > > > > > I think there is not much to add to the discussion at this point.

---

> > > > > > > > ### Author Response · Authors · 2021-11-28
> > > > > > > > **final remarks**
> > > > > > > >
> > > > > > > > We think we do see this point, we didn't simply ignore it. In fact, we fully agree with the referee that tasks from other domains, like NLP, would be highly interesting to look at from the DS perspective we developed here (or, more generally, to investigate how the theory could be exploited for other domains and use cases, as the referee suggested). At the same time we are aware, though, that this is considerable work (far) beyond just a single paper, but would constitute a development in its own right, especially since the DS literature in other (than scientific) areas is still quite underdeveloped.
> > > > > > > >
> > > > > > > > The area of DS reconstruction is a big and scientifically important and timely field in its own right, and many papers at ML conferences (incl. ICLR) deal just with new algorithms for this particular application area (see our Related Work sect.). We are sorry if our remarks about DS apparently came across the wrong way. We had the impression that this referee comes from a different area, with a different perspective, and therefore perhaps underestimates a bit the significance for scientific ML. Nothing else we had wanted to express (and thought we had done so in a non-judgmental way). And please see that already in our previous replies we did comment on the suggestion of extension to other use cases.
> > > > > > > > Beyond our scientific application case, however, we believe the points we worked out mathematically are still important to the community more generally, as are the potential directions for solutions we lined out (the points we made will still be true and relevant for other types of time series, even if we didn't work them out explicitly).
> > > > > > > >
> > > > > > > > We once again thank the referee for her/his detailed scrutiny of our paper. This way or the other, the comments certainly helped us to see more clearly which parts of the manuscript needed clarification, and the newly added experimental examples we think already broadened the scope (beyond just adding further evidence).

---

> > > > > > > > > ### Author Response · Authors · 2021-11-29
> > > > > > > > > **Confused**
> > > > > > > > >
> > > > > > > > > We are somewhat confused: The referee previously stated "In general I'm satisfied with the answers by the authors and will raise my score accordingly." But then modified the reply again, apparently because s/he was upset by a (perhaps incautious) remark we made (the first sentence of our reply further above?).
> > > > > > > > >
> > > > > > > > > Just to clarify: We didn't question the ML expertise of this referee obviously (and already apologized should our remarks have suggested otherwise; they certainly weren't meant that way). People reviewing ICLR papers come from all different directions and backgrounds (we have been ourselves referees for ICLR multiple times, including presently). We are furthermore amply aware this is one of the purposes of a proper review process, to combine different perspectives and learn from them. Our impression here was solely that the referee perhaps underestimates a bit that DS reconstruction is a larger and valid topic in its own right, and the implications for this field (and, honestly, this basically also comes through in her/his last response). We *did* apply our framework to standard use cases, *in the area of scientific ML and DS reconstruction*.
> > > > > > > > >
> > > > > > > > > We have invested two intense, busy weeks into this revision, primarily addressing this referee's points. So on this basis alone we think it is clear we took this referee's points very seriously, and we acknowledged this multiple times in the process.

---

> > > > > > > > > > ### Author Response · Authors · 2021-12-09
> > > > > > > > > > **still puzzled about one point**
> > > > > > > > > >
> > > > > > > > > > Dear Referee,
> > > > > > > > > >
> > > > > > > > > > We would like to make another attempt to come back to your major point. It appears we still may have missed something.
> > > > > > > > > >
> > > > > > > > > > Before getting there, we once again apologize that one of our remarks was apparently upsetting. It certainly wasn’t meant confrontational and, in our minds, we didn’t even speculate about the referee’s background. Rather, we had the impression that there was a deeper misunderstanding about the role of DS theory in our specific context, which we had hoped we could clarify.
> > > > > > > > > >
> > > > > > > > > > What we are still puzzled about is the following: If we understood correctly, the referee criticizes that we didn’t consider in his view more relevant use cases like sequential MNIST or NLP. But why are these considered more relevant than climate or EEG data? In our understanding, seq. MNIST is an essentially artificial dataset (at least its time series nature), while climate data constitute a challenging and highly relevant real-world problem. Same goes for many other dynamical systems, like epidemiological dynamics or other physiological time series. At a societal level, these are extremely relevant use cases in our minds, on which we had tested our theory.
> > > > > > > > > > So our question is: Is there still a point we missed here? If so, it would help us if the referee could further comment on this. Thanks.

---

### Author Response · Authors · 2021-11-18
**General reply**

*New results & modifications*

We have added a number of new examples, figures and benchmarks to our empirical section, namely:
- Results on non-autonomous dynamical systems with external time-varying inputs (new Fig. 1b, replacing the previous Rössler example).
- Results on high-dimensional chaotic systems (new Fig. 7).
- Results on partially observed dynamical systems (new Fig. 8).
- Results on other training procedures (new Fig. 9), highlighting that a mere conventional windowing/batching approach is not sufficient.

We have furthermore modified our text to
- clear up the profound misunderstanding that our theory is limited to special cases only, without external forcing (in fact it is general and covers *all* known scenarios, *incl.* those with external inputs, as now also illustrated in new Fig. 1b),
- include the suggested references and address all referee points as specified in our detailed replies below.


*Limited scope of theory/ external inputs*

There is a profound and very unfortunate misunderstanding here, which we are aware we have largely caused ourselves: Our theory is completely general, it covers *all* theoretically possible dynamical scenarios, *with* or without external inputs. None of our theorems assumes the dynamics to be autonomous, all of the theory applies to non-autonomous (externally forced; eq. 1) systems just as well (made very explicit in Remark 2 now).

We further would like to clarify that with ‘no inputs’ we mainly meant that the problem in scientific scenarios is usually treated as an *unsupervised* one. This means that rather than feeding the observed sequences directly as inputs to the RNN (as in NLP), we treat them as *observations*, as the aim of RNN training here is to approximate the underlying DS that generated the observations (as illustrated in Figs. 1-3). In fact, from a theoretical perspective it actually makes no difference whether we include any variables as explicit external inputs or as dynamical variables in their own right, i.e. treat them as observations, since, mathematically, non-autonomous (externally forced) systems can always be rewritten as autonomous dynamical systems (e.g. Zhang et al. 2009; Perko 2001; Alligood et al. 1996). For instance, note that our approach also worked for the real temperature data, where we may assume external inputs but have no explicit knowledge of them. This also makes clear that the exact dynamics does not need to be known in advance, beyond what we can extract through well established methods as those we have used here (cf. Figs. 3a, 4b).

To fully settle this point, we now have also added an example of an explicitly externally forced system to the experimental evaluation section (new Fig. 1b).

---

> ### Author Response · Authors · 2021-11-18
> **General reply (cont'd)**
>
> *Empirical evaluation in a theoretical paper*
>
> This work is a theoretical study, and as far as we can see, it’s the first of its kind. It introduces a number of novel theorems, accompanied by comprehensive proofs that partly draw on advanced concepts in topology and ergodic theory. It provides a comprehensive mathematical coverage of the relations between RNN dynamics, under all theoretically possible scenarios, and the behavior of the error gradients, including highly important cases like chaos or quasi-periodicity that have not been addressed in the literature before. It exposes formally important constraints that need to be acknowledged by any training algorithm or architecture if the underlying time series are chaotic, that have not been pointed out so far. This is not just another take on the EVGP, but a profoundly novel theoretical contribution with fundamental implications (at least for the problem of DS reconstruction) that we have not seen being considered so explicitly elsewhere. We feel this theoretical contribution got somehow lost in the process and was quite undervalued by some of the reviews which mainly focused on the empirical evaluation.
>
> While we now have added many further examples and evaluations (see list above) that further highlight the empirical relevance of our theory, we would like to stress that the empirical section was added mainly to illustrate the implications of the theory (initially we had even planned to submit only the theoretical part as it is a comprehensive body of work in its own right). Because the theory naturally builds on the concept of Lyapunov exponents which is so central to dynamical systems, it was the job of the empirical section to illustrate the implications of the system’s max. Lyapunov exponent for RNN training. We neither aimed for a SOTA here, nor did we pretend that the training algorithm itself is completely novel (although our specific, theoretically motivated implementation is indeed new and made a decisive difference in training on DS). Rather the purpose of the empirical section was to demonstrate general implications of our theory for any training scheme or architecture that need to be considered. We confirmed this for a variety of different architectures (vanilla RNN, PLRNN, LSTM) and training procedures (gradient clipping, teacher forcing), and now added further examples (new Figs. 1b, 7-9) to the revision. We worked this out on examples that are most relevant to science, namely dynamical systems reconstruction where one often just has a set of observed time series from which one likes to reconstruct the underlying system. This is a huge and important application area in its own right (see our Related Works sect.).
>
> *References*
>
> Kathleen T. Alligood, Tim D. Sauer, and James A, Yorke.Chaos:  An Introduction to DynamicalSystems. Springer, New York, NY, 1996.
> Ludwig Arnold, Random Dynamical Systems. Springer, 1991.
> Boris  P.  Bezruchko, Dmitry  A.  Smirnov, Extracting  Knowledge  From  Time  Series: An  Introduction to Nonlinear Empirical Modeling, Springer Series in Synergetics, Springer-Verlag, 2010.
> Bo Chang, Minmin Chen, Eldad Haber, Ed H. Chi, AntisymmetricRNN: A Dynamical System View on Recurrent Neural Networks, ICLR, 2019.
> Ken-ichi, Funahashi, Yuichi, Nakamura, Approximation of dynamical systems by continuous time recurrent neural networks, Neural Networks, 6(6):801-806, 1993.
> John Guckenheimer, Philip Holmes, Nonlinear Oscillations, Dynamical Systems, and Bifurcations of Vector Fields, Springer, New York, NY, 1983.
> Joshua Hanson, Maxim Raginsky, Universal Simulation of Stable Dynamical Systems by Recurrent Neural Nets, PMLR, 120:384-392, 2020.
> Holger Kantz, Thomas Schreiber. Nonlinear time series analysis. Cambrdige, 2003.
> Masahiro Kimura, Ryohei Nakano, Learning dynamical systems by recurrent neural networks from orbits, Neural Networks,11(9):1589-1599, 1998.
> Edward Ott, Chaos in Dynamical Systems, 2nd Edition, Cambridge University Press, 2002.
> Lawrence Perko, Differential Equations and Dynamical Systems, volume 7.  Springer, New York,NY, 2001.
> T. Konstantin Rusch, Siddhartha Mishra, Coupled oscillatory recurrent neural network (coRNN): An accurate and (gradient) stable architecture for learning long time dependencies, ICLR, 2021a.
> T. Konstantin Rusch, Siddhartha Mishra, Unicornn: A recurrent model for learning very long time dependencies, In Marina Meila and Tong Zhang (eds.), PMLR, 18-24, 2021b.
> Ryan Vogt, Maximilian Puelma Touzel, Eli Shlizerman, Guillaume Lajoie, On Lyapunov Exponents for RNNs: Understanding Information Propagation Using Dynamical Systems Tools, arXiv:2006.14123, 2020.
> Huaguang Zhang, Derong Liu, and Zhiliang Wang, Controlling Chaos. Springer, London, 2009.

---

### Author Response · Authors · 2021-11-23
**UPDATE**

In addition to our previous updates (examples with explicit external forcing, high-dimensional systems, partially observed systems, other training procedure; see list below), we now have furthermore added another real-world dataset (electrophysiological recordings, new Figs. 10,11). This example further confirms that empirically estimated Lyapunov exponents provide a sensible guidance to optimal reconstruction (despite the fact that in empirical situations we usually don’t have explicit knowledge about all external inputs, confirming the theoretical points we made below in our response).

We would like to thank the referees for their scrutiny of our work, and for pushing us to work out a number of points much more clearly. Initially, we were admittedly somewhat frustrated that apparently we didn’t manage to get across major points of our work, especially the implications of our theoretical part. But we hope the additional experimental evaluation, text changes and our responses below will clarify these issues.

---

### Decision · Program_Chairs · 2022-01-20

**Decision:**

Reject

**Comment:**

This paper analyzes the gradient behavior of RNNs in terms of the Lyaponuv exponents of its trajectory/orbit, showing that RNNs with cyclic or stable equilibrium dynamics have bounded gradients, but if the dynamics are chaotic the gradients will explode. From these insights, the authors propose an algorithmic remedy for this pathology, which is essentially a teacher-forcing method that periodically projects the observation onto the hidden state during training. A thorough empirical investigation is performed showing the utility of the proposed approach for modeling chaotic data.

The reviewers had split opinions on this paper. Some reviewers found value in the theoretical contributions and the connection between Lyapunov exponents and behaviors of the dynamics of recurrent neural networks, while others thought the theoretical framework may have limited practical utility. Several reviewers found the initial experiments to be lacking, though many of their concerns were alleviated after the substantial additions the authors provided during the discussion phase.

I believe the observation that exploding gradients are unavoidable when modeling chaotic data is important and would be of significant interest to the broader ICLR community. However, the practical implications of this observation have not been thoroughly described or investigated, and without this perspective, the theoretical results by themselves are much less impactful. In practice, it is usually the case that the ground-truth function is not learned exactly, the time horizon is finite, the gradients are noisy, the data-generating process is opaque, etc. Do these caveats have any bearing on the conclusions? The experiments address some of these questions, but only indirectly, and a more explicit discussion of the practical implications would broaden the impact of the paper.

Along the same lines, the practical utility of the theoretical framework could be further supported if there were some analysis of more varied or additional RNN use-cases. As one reviewer mentioned, I think the ICLR community in particular would appreciate any theoretical or algorithmic insights that might yield improvements on a standard baseline task like seqMNIST, which has served as a point-of-comparison for many alternative methods and which would facilitate comparisons to prior work.

Overall, this paper does make some nice and potentially important theoretical insights about training RNNs on chaotic data, and it does include an extensive battery of empirical evaluations, however the practical implications remain largely unconvincing, and I believe the paper falls just short of the bar for acceptance.

---

> ### Public Comment · ~Daniel_Durstewitz1 · 2022-02-05
> **Public reply to decision note**
>
> We thank the AC very much for her/his very considerate, fair, and detailed feedback and summary, and also for acknowledging the strong points about our work.
>
> We would like to strongly object, however, to the conclusion that our work may be of limited practical relevance, or that we would not have shown its practical relevance sufficiently. This assessment by the referees was based on the initial misunderstanding that our results were only true for autonomous systems (which we feel would have been important enough, as it is actually quite a common situation in dynamical systems reconstruction where one only has observed time series but doesn't know the inputs to the system). In our revision, however, we had *thoroughly* addressed this point both mathematically and with additional experimental examples. Unfortunately, three of the referees didn't get back to us at all after our extensive revision, while the fourth referee seemed initially convinced, but then suddenly changed his/her mind and deleted his/her previous affirmative response, as s/he apparently was angered by what we thought was a rather innocent remark in our rebuttal.
>
> While for the specific area of NLP the implications of our results may require further discussion, we have amply illustrated their implications for scientific machine learning and the natural sciences with many examples. The Lorenz or Roessler attractor, or the forced Duffing oscillator, for instance, are very common and popular benchmarks in this field of dynamical systems reconstruction. The practical issues mentioned (finite time series, noisy gradients, partial observations, only approximations to ground-truth etc.) *all* occur in our examples, especially in the real-world data of course. In this area, we also consider climate or medical/physiological time series practically far more relevant than seqMNIST (the example brought up by the fourth referee): seqMNIST is a purely artificial problem (on which, on top, to our experience, most advanced RNNs achieve close to 100% performance, i.e., it doesn’t really discriminate that well). It’s not even clear whether seqMNIST bears signatures of chaos and hence whether it’s a relevant test case for our theory at all.
>
> Perhaps these are points that were not clear enough in our first submission, but we thought we had extensively clarified this during the review process and in the revised version (together with the original major criticism that our approach would neglect external inputs, which is something we had addressed at length).